# Large-scale features of Last Interglacial climate: Results from evaluating the *lig127k* simulations for CMIP6-PMIP4

Bette L. Otto-Bliesner[1], Esther C. Brady[1], Anni Zhao[2], Chris Brierley[2], Yarrow Axford[3], Emilie Capron[4], Aline Govin[5], Jeremy Hoffman[6,7], Elizabeth Isaacs[2], Masa Kageyama[5], Paolo Scussolini[8], Polychronis C. Tzedakis[2], Charlie Williams[9], Eric Wolff[10], Ayako Abe-Ouchi[11], Pascale Braconnot[5], Silvana Ramos Buarque[12], Jian Cao[13], Anne de Vernal[14], Maria Vittoria Guarino[15], Chuncheng Guo[16], Allegra N. LeGrande[17], Gerrit Lohmann[18], Katrin J. Meissner[19], Laurie Menviel[19], Polina A. Morozova[20], Kerim H. Nisancioglu[21,22], Ryouta O'ishi[11], David Salas Y Mélia[12], Xiaoxu Shi[18], Marie Sicard[5], Louise Sime[15], Christian Stepanek[18], Robert Tomas[1], Evgeny Volodin[23], Nicholas K.H. Yeung[19], Qiong Zhang[24], Zhongshi Zhang[16,25], Weipeng Zheng[26]

[1]Climate and Global Dynamics Laboratory, National Center for Atmospheric Research, Boulder, 80305, USA
[2]Environmental Change Research Centre, Department of Geography, University College London, London WC1E 6BT, UK
[3]Department of Earth & Planetary Sciences, Northwestern University, Illinois, USA
[4]Physics of Ice, Climate and Earth, Niels Bohr Institute, University of Copenhagen, Copenhagen, 2200, Denmark.
[5]LSCE-IPSL, Laboratoire des Sciences du Climat et de l'Environnement (CEA-CNRS-UVSQ), University Paris-Saclay, Gif sur Yvette, 91190, France
[6]Science Museum of Virginia, Richmond, Virginia, 23220, USA
[7]Center for Environmental Studies, Virginia Commonwealth University, Richmond, VA, 23220, USA
[8]Institute for Environmental Studies, Vrije Universiteit Amsterdam, Amsterdam, The Netherlands
[9]School of Geographical Sciences, University of Bristol, Bristol, UK
[10]Department of Earth Sciences, University of Cambridge, Cambridge, CB2 3EQ, United Kingdom.
[11]Atmosphere and Ocean Research Institute, University of Tokyo, 5-1-5, Kashiwanoha, Kashiwa-shi, Chiba 277-8564, Japan
[12]CNRM (Centre National de Recherches Météorologiques), Université de Toulouse, Météo-France, CNRS (Centre National de la Recherche Scientifique), Toulouse, France
[13]Earth System Modeling Center, Nanjing University of Information Science and Technology, Nanjing, 210044, China
[14]Departement des sciences de la Terre et de l'atmosphere, Universite du Quebec, Montreal, Canada
[15]British Antarctic Survey, High Cross, Madingley Road, Cambridge, CB3 0ET, UK
[16]NORCE Norwegian Research Centre, Bjerknes Centre for Climate Research, Bergen 5007, Norway
[17]NASA Goddard Institute for Space Studies and Center for Climate Systems Research, Columbia University, New York City, USA
[18]Alfred Wegener Institute - Helmholtz Centre for Polar and Marine Research, Bussestr. 24, 27570 Bremerhaven, Germany
[19]Climate Change Research Centre, ARC Centre of Excellence for Climate Extremes, The University of New South Wales, Sydney, NSW 2052, Australia
[20]Institute of Geography, Russian Academy of Sciences, Staromonetny L. 29, Moscow, 119017, Russia
[21]Department of Earth Science, University of Bergen, Bjerknes Centre for Climate Research, Allégaten 41, 5007, Bergen, Norway
[22]Centre for Earth Evolution and Dynamics, University of Oslo, Oslo, Norway
[23]Marchuk Institute of Numerical Mathematics, Russian Academy of Sciences, 119333 Moscow, Russia
[24]Department of Physical Geography and Bolin Centre for Climate Research, Stockholm University, Stockholm, 10691, Sweden
[25]Department of Atmospheric Science, School of Environmental studies, China University of Geoscience, Wuhan, 430074, China
[26]LASG (State Key Laboratory of Numerical Modeling for Atmospheric Sciences and Geophysical Fluid Dynamics), Institute of Atmospheric Physics, Chinese Academy of Sciences, Beijing 100029, China

*Correspondence to*: Bette Otto-Bliesner (ottobli@ucar.edu)

**Abstract.** The modeling of paleoclimate, using physically based tools, is increasingly seen as a strong out-of-sample test of the models that are used for the projection of future climate changes. New to CMIP6 is the Tier 1 Last Interglacial experiment for 127 thousand years ago (*lig127k*), designed to address the climate responses to stronger orbital forcing than the *midHolocene* experiment, using the same state-of-the-art models as for future and following a common experimental protocol. Here we present a first analysis of a multi-model ensemble of 17 climate models, all of which have completed the CMIP6 DECK experiments. The Equilibrium Climate Sensitivity (ECS) of these models varies from 1.8 to 5.6°C. The seasonal character of the insolation anomalies results in strong summer warming over the Northern Hemisphere continents in the *lig127k* ensemble as compared to the CMIP6 *piControl* and much-reduced minimum sea ice in the Arctic. The multi-model results indicate enhanced summer monsoonal precipitation in the Northern Hemisphere and reductions in the Southern Hemisphere. These responses are greater in the *lig127k* than the CMIP6 *midHolocene* simulations as expected from the larger insolation anomalies at 127 ka than 6 ka.

New syntheses for surface temperature and precipitation, targeted for 127 ka, have been developed for comparison to the multi-model ensemble. The *lig127k* model ensemble and data reconstructions are in good agreement for summer temperature anomalies over Canada, Scandinavia, and the North Atlantic, and for precipitation over the Northern Hemisphere continents. The model-data comparisons and mismatches point to further study of the sensitivity of the simulations to uncertainties in the boundary conditions and of the uncertainties and sparse coverage in current proxy reconstructions.

The CMIP6-PMIP4 *lig127k* simulations, in combination with the proxy record, improve our confidence in future projections of monsoons, surface temperature, and Arctic sea ice, thus providing a key target for model evaluation and optimization.

## 1    Introduction

Quaternary interglacials can be thought of as natural experiments to study the response of the climate system to variations in forcings and feedbacks (Tzedakis et al., 2009). The current interglacial (Holocene, the last 11,600 years) and the Last Interglacial (LIG; ~129,000-116,000 years before present) are well represented in the geological record and provide an opportunity to study the impact of differences in orbital forcing. Two interglacial timeslices, the mid-Holocene (*midHolocene* or MH, ~6,000 years before present) and the early part of the LIG (*lig127k*; 127,000 thousand years before present), are included as Tier 1 simulations in the Coupled Model Intercomparison Project (CMIP6) and Paleoclimate Modeling Intercomparison Project (PMIP4). These equilibrium simulations are designed to examine the impact of changes in the Earth's orbit and hence the latitudinal and seasonal distribution of incoming solar radiation (insolation) at times when atmospheric greenhouse gas levels and continental configurations were similar to those of the preindustrial period. They test our understanding of the interplay between radiative forcing and atmospheric circulation, and the connections between large-scale and regional climate changes giving rise to phenomena such high-latitude amplification in temperature changes, and responses of the monsoons, as compared to today.

The modeling of paleoclimate, using physically based tools, has long been used to understand and explain past environmental and climate changes (Kutzbach and Otto-Bliesner, 1982; Braconnot et al., 2012; Harrison et al., 2015; Schmidt et al., 2014). In the first phase of PMIP, the MH and the Last Glacial Maximum (LGM, ~21,000 years ago) were identified as important time periods to compare data reconstructions and model simulations (Joussaume et al., 1999; Braconnot et al., 2000). A novel aspect in CMIP5 was applying the same models and configurations used in the

paleoclimate simulations as in the transient 20th century and future simulations, providing consistency – both in the overall forcings and in how they are imposed – between experiments. In addition to MH and LGM experiments, CMIP5 and PMIP3 included coordinated protocols for Last Millennium (LM, 850-1850 CE) and the mid-Pliocene Warm Period (mPWP, 3.3-3.0 million years ago) experiments.

The LIG is recognized as an important period for testing our knowledge of climate and climate-ice sheet interactions to forcing in warm climate states. Although the LIG was discussed in the First Assessment Report of the IPCC (Folland et al., 1990), it gained more prominence in the IPCC Fourth and Fifth Assessments (AR4 and AR5) (Jansen et al., 2007; Masson-Delmotte et al., 2013). Evidence in the geologic record indicate a warm Arctic (CAPE, 2006; Turney et al., 2010) and a global mean sea level highstand at least 5 m higher (but probably no more than 10 m

higher) than present for several thousand years during the LIG (Dutton et al., 2015). The ensemble of LIG simulations examined in the AR5 (Masson-Delmotte et al., 2013) was not wholly consistent; the orbital forcing and GHG concentrations varied between the simulations. While it had been suggested that differences in regional temperatures between models might reflect differences in cryosphere feedback strength (Yin and Berger, 2012; Otto-Bliesner et al., 2013) or differences in the simulation of the Atlantic Meridional Overturning Circulation (AMOC)

(Bakker et al., 2013), differences between models could also have arisen because of differences in the experimental protocols. Furthermore, the LIG simulations were mostly made with older and/or lower-resolution versions of the models than were used for future projections, making it more difficult to use the results to assess model reliability (Lunt et al., 2013).

For the first time a LIG experiment is included as a CMIP6 simulation, setting a common experimental protocol and asking modeling groups to run with the same model and at the same resolution as the DECK simulations (Otto-Bliesner et al., 2017). At the PAGES QUIGS workshop in Cambridge in 2015, the community identified the 127 ka time slice for the CMIP6-PMIP4 LIG experiment for several reasons: large Northern Hemisphere seasonal insolation anomalies, no (or little) remnants of the North American and Eurasian ice sheets, and sufficient time to allow for

dating uncertainties to minimize the imprint of the previous deglaciation and the Heinrich 11 (H11) meltwater event (Marino et al., 2015). The Tier 1 *lig127k* experiment addresses the climate responses to stronger orbital forcing, relative to the *midHolocene*. It also provides a basis to address the linkages between ice sheets and climate change in collaboration with the Ice Sheet Model Intercomparison Project for CMIP6 (ISMIP6) (Nowicki et al., 2016).

In this paper, we start with a brief overview of the experimental design of the *lig127k* (Otto-Bliesner et al., 2017). We briefly summarize the simulation of temperature, precipitation, and sea ice, in the subset of CMIP6 *piControl*

simulations that have corresponding *lig127k* simulation, as compared to observational datasets. We then provide an initial analysis of the multi-model ensemble mean and model spread in the *lig127k* surface temperature, precipitation, and Arctic sea ice responses as compared to the CMIP6 DECK *piControl* simulations. A new syntheses of surface temperature and precipitation proxies, targeted for 127 ka, is used for comparison to the model simulations. We also explore differences in the responses of surface temperature, monsoon precipitation, and Arctic sea ice to the different magnitudes and seasonal character of the insolation anomalies at 127 ka versus 6 ka. We then conclude with a discussion of possible reasons for the model-data differences and implications for future projections.

## 2    Methods

### 2.1    Experimental design

The CMIP DECK *piControl* for 1850 CE (see Eyring et al. 2016 for description of this experiment) is the reference simulation to which the *lig127k* paleo-experiment is compared. The modeling groups were asked to use the same model components and follow the same protocols for implementing external forcings as used in the *piControl*. The boundary conditions for the *lig127k* and *piControl* experiments are described in Otto-Bliesner et al. (2017) and the Earth System Documentation (2019). More detailed information is given below and in Table 1.

Earth's orbital parameters (eccentricity, longitude of perihelion, and obliquity) are prescribed following Berger and Loutre (1991). The DECK *piControl* simulations use the orbital parameters appropriate for 1850 CE (Table 1, Fig. 1) (Eyring et al., 2016), when perihelion occurs close to the boreal winter solstice. The orbit at 127 ka is characterized by larger eccentricity than at 1850 CE, with perihelion occurring close to the boreal summer solstice (Table 1, Fig. 1). The tilt of the Earth's axis was maximal at 131 ka and remained higher than in 1850 CE through 125 ka; obliquity at 127 ka was 24.04° (Table 1). The solar constant for the *lig127k* simulations is prescribed to be the same as in the DECK *piControl* simulation.

The orbital parameters affect the seasonal and latitudinal distribution and magnitude of solar energy received at the top of the atmosphere, resulting in large positive insolation anomalies during boreal summer at 127 ka as compared to 1850 CE (Fig. 1). Positive insolation anomalies are present from April to September and from 60°S to 90°N. These anomalies peak at over 70 W m$^{-2}$ in June at 90°N. Insolation in the Arctic (defined here as 60-90°N) is more than 10% greater at 127 ka than 1850 CE during May through early August. The higher obliquity at 127 ka contributes to a small but positive annual insolation anomaly compared to preindustrial at high latitudes in both hemispheres and negative annual insolation anomaly at tropical latitudes. The global difference in annual insolation forcing between the *lig127k* and *piControl* experiments is negligible.

Ice-core records from Antarctica provide measurements of the well-mixed GHGs: $CO_2$, $CH_4$, and $N_2O$. By 127 ka, the concentrations of atmospheric $CO_2$ and $CH_4$ had increased from their minimum levels during the previous glacial period to values comparable to, albeit somewhat lower than, preindustrial levels (Table 1).

Natural aerosols show large variations on glacial-interglacial time scales, with low aerosol loadings during interglacials compared to glacials, and during the peak of the interglacials compared to present day (Albani et al., 2015; deMenocal et al., 2000; Kohfeld and Harrison, 2000). Modeling groups were asked to implement changes in atmospheric dust aerosol in their *lig127k* simulations following the treatment used for their DECK *piControl* simulations (see Table 2 for details). The background volcanic stratospheric aerosol used in the CMIP6 DECK *piControl* was also to be used for the *lig127k* simulation. Other aerosols included in the DECK *piControl* simulations should similarly be included in the *lig127k* simulations.

There is evidence for changes in vegetation distribution during the LIG (e.g. LIGA Members, 1991; CAPE, 2006; Larrasoana, 2013). However, there is insufficient data coverage for many regions to be able to produce reliable global vegetation maps. Furthermore, given the very different levels of complexity in the treatment of vegetation properties in the current generation of climate models, paleo-observations do not provide sufficient information to constrain their behavior in a comparable way. The treatment of natural vegetation in the *lig127k* simulations was therefore to be the same as in the DECK *piControl* simulation. Accordingly, depending on what was done in the DECK *piControl* simulation, vegetation could either be prescribed to be the same as in that simulation, prescribed but with interactive phenology, or predicted dynamically (see Table 2 for implementations in the models).

Paleogeography and ice sheets were to be kept at their present-day configuration.

## 2.2 Model evaluation

The 17 modeling groups that have completed CMIP6 *lig127k* simulations are presented in this paper (Table 2). All used the CMIP6 version of their model also used for their DECK experiments. The Equilibrium Climate Sensitivity (ECS) varies from 1.8 to 5.6°C. The years analyzed for each model and DOIs for each of the simulations are given in Supplementary Table S1. The analysis uses data available on the CMIP6 ESGF for surface air temperature (tas), precipitation (pr), and sea ice concentration (siconc).

## 2.3 Calendar adjustments

The output is corrected following Bartlein and Shafer (2019), to account for the impact that the changes in the length of months or seasons over time have on the analysis (Fig. 1). This correction is necessary to account for the impact of the changes in the eccentricity of the Earth's orbit and the precession when using the 'celestial' calendar. Not considering the "paleo-calendar effect" can prevent the correct interpretation of data and model comparisons at 127 ka, with the largest problems occurring in boreal fall/austral spring (Joussaume and Braconnot, 1997; Bartlein and Shafer, 2019). A more detailed discussion of the application of the PaleoCalAdjust software to past time periods with strong orbital forcing can be found in Bartlein and Shafer (2019) and Brierley et al. (2020).

## 3   Simulation results

## 3.1 Preindustrial simulations

Brierley et al. (2020) provide an extensive evaluation of the CMIP6 preindustrial simulations as compared to observational datasets: reanalyzed climatological temperatures (between 1871-1900 CE; Compo et al., 2011) for the spatial patterns, zonal averages of observed temperature for the period 1850-1900 CE from the HadCRUT4 dataset (Morice et al., 2012; Ilyas et al., 2017), and climatological precipitation data for the period between 1970 and the present day (Adler et al., 2003). In summary, they find that the PMIP4-CMIP6 models are in general cooler than the observations, most noticeably at the poles, over land and the NH oceans. The poleward extent of the North African monsoon, in particular, is underestimated in the CMIP6 preindustrial simulations.

The CMIP6 *midHolocene* and *lig127k* have 14 models in common (see Fig 4a in Brierley et al. (2020) and Fig. 2a in this paper). The *piControl* multi-model mean (MMM), zonal-average temperature is slightly cooler than observed at high (60-90°N) Northern Hemisphere (NH) latitudes (Fig. 2a). There is a large spread across the models though, with 8 of the models simulating colder (up to 4°C) than observed temperatures, and 9 of the models simulating warmer (up to 2°C) then observed temperatures. The *piControl* MMM, zonal-average temperature is noticeably warmer than observed at high (60-90°S) Southern Hemisphere (SH) latitudes, again with a large spread across the models. Two models, MIROC-ES2L and EC-Earth3-LR have biases in excess of 5°C. Hajima et al. (2020) attribute the MIROC-ES2L *piControl* warm bias over the Southern Ocean to be mainly associated with the model's representation of cloud radiative processes. The spread of the *piControl* simulations is smaller at low and mid-latitudes (Fig. 2a).

We adopt the definition of sea-ice area of the Sea Ice Model Intercomparison Project (SIMIP, SIMIP Community, 2020), i.e. sea ice concentration times the cell area. The multi-model ensemble of *piControl* simulations of minimum (August-September) Arctic sea ice distribution (Fig. 3a, S2), show good agreement with the 15% contour from the HadISST data averaged over the 1870-1920 period (Fig. S1), (Rayner et al. 2003). Two models, FGOALS-g3 and EC-Earth3-LR, show noticeably greater minimum summer sea ice extent in the Nordic Seas as compared to the HadISST period (Fig. S2). Further, evaluation of the *piControl* simulations can be found in Kageyama et al. (2020). In particular, they find that in comparison to sea ice reconstruction sites, the models generally overestimated sea ice cover at sites close to the sea ice edge.

Figure 4 shows the seasonal cycle of Arctic sea ice area in the *piControl* simulations for each model and the MMM. These are compared to the NOAA OI_v2 observational dataset, with higher temporal and spatial coverage than the HadISST dataset. The NOAA_OI_v2 dataset (Reynolds et al., 2003), also used in Kageyama et al. (2020), only extends back to 1981. It should be noted that atmospheric $CO_2$ concentrations had already risen to 340 ppm by 1981, as compared to 284.7 ppm specified in the *piControl* simulations. We find a large spread across the *piControl* simulations. The range in March is 12.27 to 19.16 mill $km^2$ and the MMM is 15.30 ± 1.89 mill $km^2$. The range in September is 3.56 to 9.73 mill $km^2$ and the MMM is 6.13 ± 1.66 mill $km^2$. Generally, those models with less sea ice in March than the MMM also have less sea ice in September than the MMM. Observed estimates of sea ice area from the NOAA-OI_v2 dataset for 1982 -2001 are 14.7 mill $km^2$ for March and 5.1 mill $km^2$ for September.

The MMM *piControl* simulations of austral summer minimum (February-March) sea ice distribution around Antarctica however, shows less consensus among the models and less agreement with the HadISST data, with many models significantly underestimating the observed austral summer minimal extent (Figs. 3b, 4b, S4). The range in February is 0.02 to 3.82 mill km$^2$ and the MMM is 1.65 ± 1.21 mill km$^2$. Antarctic sea ice melts back largely to the continent's edge in February-March in four models (AWI-ESM-2-1-LR, EC-Earth3-LR, MIROC-ES2L, and MPI-ESM1-2-LR) (Fig. S5). The spread of models is even greater in their simulations of *piControl* austral winter sea ice area around Antarctica, ranging from 3.27 mill km$^2$ in September in MIROC-ES2L to over 19 mill km$^2$ in IPSL-CM6-LR and FGOALS-g3. The September MMM is 17.13 ± 5.21 mill km$^2$. Observed estimates of sea ice area from the NOAA-OI_v2 dataset for 1982 -2001 are 2.7 mill km$^2$ for February and 16.5 mill km$^2$ for September.

### 3.2 Surface temperature responses

The seasonal character of the insolation anomalies results in warming and cooling over the  continents in the *lig127k* ensemble (relative to the *piControl*) in JJA and DJF, respectively, except for the African and southeast Asian monsoon regions in JJA. These patterns of seasonal, continental warming and cooling are a robust feature across the models, with more than 70% of the models agreeing on the sign of the temperature change (Fig. 5a,c).

The warming during JJA is greater than 6°C at mid-latitudes in North America and Eurasia (Fig. 5a), though with significant differences in the magnitude of the warming in southeast U.S, Europe, and eastern Asia among the models (Fig. 5b). Further investigation of the effects of preindustrial vegetation, including crops, for these regions in the lig127k protocol would be useful (Otto-Bliesner et al., 2020). Subtropical land areas in the Southern Hemisphere (SH) also respond to the positive (but more muted) insolation anomalies, with JJA temperature anomalies more than 2°C warmer than PI. JJA warming over most of the oceans is a robust feature across the models. This warming is greatest in the North Atlantic and the Southern Ocean, though with large differences across the ensemble of models (Fig. 5b). Cooling over the Sahel and southern India in JJA is associated with the increased cloud cover associated with the enhanced monsoons (see Section 3.4).

In response to the negative insolation anomalies at all latitudes (Fig. 1), the *lig127k* MMM simulates cooling during DJF over the continental regions of both hemispheres and low and mid-latitude oceans (Fig. 5c). The largest DJF temperature anomalies occur over southeastern Asia and northern Africa. Ocean memory has been shown to provide the feedback to maintain positive or neutral DJF temperature anomalies in the Arctic and North Atlantic (see Serreze and Barry (2011) for further discussion). As indicated by the standard deviations of the ensemble changes, large differences in the magnitude of the DJF high-latitude, surface temperature responses and feedbacks, exist among the models (Fig. 5d). Understanding these differences warrants further analyses in future studies.

These seasonal patterns of change are similar to those found in Lunt et al. (2013), though the warming is larger in the CMIP6 simulations. It should be noted that the MMM in Lunt et al. includes simulations that have varying orbital years (between 125 ka and 130 ka) and greenhouse-gas concentrations.

Annually, the MMM surface temperature changes between the *lig127k* and *piControl* are generally less than 1°C over most of the globe, with two exceptions (Fig. 5e): greater negative surface temperature anomalies across the North African and Indian monsoon regions, and positive surface temperature anomalies in the Arctic. Although more than 70% of the models agree on the sign of the changes in these regions, as well as in the Indian sector of the Southern Ocean (Fig. 5e), the across ensemble standard deviations indicate differences in the magnitudes of the annual surface temperature responses (Fig. 5f). Globally, the MMM change in annual surface air temperature is close to zero (-0.2 ± 0.32°C), though with a large spread among the models (-0.48 to 0.56°C) (Table 3). Conclusions about the land versus ocean or NH versus SH annual temperatures changes are complicated by mean changes being close to zero and not consistently positive or negative (Table 3).

The large spread of mean annual surface temperature change among the models in the polar regions (60-90° latitude) is further illustrated in Figure 2b. Annual Arctic surface temperature changes in the *lig127k* simulations range from -0.39 to 3.88°C. The MMM is 0.82 ± 1.20°C. Notably, EC-Earth3-LR and HadGEM3-GC3.1-LL have anomalies greater than 3°C in their *lig127k* simulations as compared to their *piControl* simulations, while AWI-ESM-1-1-LR and FGOALS-f3-L are cooler in their *lig127k* simulations as compared to their *piControl* simulations. The spread (and magnitude) of mean annual temperature change for the SH polar region is less, with 7 of 17 models simulating a modest warming of 0-1°C, and three models simulating a cooling of the mean annual surface temperature (Fig. 2b). The MMM is 0.38 ± 0.63°C. The change in the NH latitudinal gradient is positive from all models, 1.27°C in the MMM though ranging quite significantly among models for 0.30°C in FGOALS-f3-L and 3.94°C in EC-Earth3-LR (Table 3). The change in the SH latitudinal gradient is smaller, 0.47°C in the MMM, reflecting the prescription of a modern Antarctic ice sheet in the *lig127k* experiment (Table 3). Changes in the size of the Antarctic ice sheet during the Last Interglacial would be expected to result in warming at polar latitudes in the SH and an increase in the SH latitudinal gradient (Bradley et al., 2012; Otto-Bliesner et al., 2013; Stone et al., 2016)

### 3.3 Sea ice responses

Boreal insolation anomalies at 127 ka enhance the seasonal cycle of Arctic sea ice (Fig. 4c). There is a ~50% reduction and shift of minimum area in the MMM from 6.1 million $km^2$ in August-September for PI to 3.1 million $km^2$ in September for *lig127k*, with a range of 0.22 to 7.47 mill $km^2$ in the individual *lig127k* simulations. The *lig127k* MMM maximum winter sea ice area in the Arctic in March is 15.68 ± 2.08 mill $km^2$ with a range of 12.27 to 20.28 mill $km^2$.The INM-CM4-8 and AWI-ESM2-1-LR have small reductions in sea ice area in all seasons with the largest decrease in October (Fig. 4e). HadGEM3-G31-ll and EC-Earth3-LR have large reductions in minimum Arctic sea ice area. HadGEM3-GC31-ll simulates an ice-free Arctic in August-September-October, with the largest reduction in October (Fig. 4c, e). EC-Earth3-LR has the largest reduction of March sea ice area for *lig127k* as compared to its *piControl,* and AWI-ESM2-1-LR has a notable increase (Fig. 4e). As shown also in Kageyama et al. (2020), PI biases in simulation of the minimum Arctic sea ice are not always a good predictor of reductions at *lig127k* (Fig. 4c).


The individual model *lig127k* minimum (August-September) Arctic sea ice area anomalies show negative correlations (-0.65) with the Arctic (60-90°N) annual surface temperature anomalies from their respective *piControl* simulations, and negative correlation (-0.53) with the corresponding JJA temperature anomalies, both significant at the 0.05 significance level (Fig. 7). Memory in the ocean and cryosphere memory provide feedbacks to maintain

positive temperature anomalies, DJF and annually, in the Arctic than in JJA (Fig. 5). Analyzing the summer atmospheric heat budgets across the models, Kageyama et al. (2020) find that the different arctic sea ice responses can be related to the sea-ice albedo feedback, i.e. phasing of the downward solar insolation changes associated with the orbital forcing and reflected upward shortwave flux changes associated with the sea-ice cover changes. As has been done for evaluating simulations of present sea ice distributions, it would be useful for further studies to also

explore model differences in the simulated changes in high-latitude cloudiness, boundary layer, winds, and ocean processes (Kattsov and Källén, 2005; Arzel et al., 2006; Chapman and Walsh, 2007).

Previous studies suggest that the mean-ice state in the control climate can influence the magnitude and spatial distribution of warming in the Arctic in future projections (Holland and Bitz, 2003). Thinner Arctic sea ice is more

susceptible to summer melting than thicker Arctic sea ice. Arctic sea ice thickness varies substantially across the 1850 CE ensemble, ranging from 1-1.5m in CMRM-CM6-1 and NESM3 to ~7.5m in MIROC-ES2L (not shown). No robust relationship to the August-September *lig127k* minimum Arctic sea ice area anomaly is present. This is also true for the CMIP6/PMIP4 *mid Holocene* simulations (Brierley et al., 2020). One reason for a lack of any relationship may be the seasonal nature of the *lig127k* and *midHolocene* insolation forcings as compared to the annual forcing by

greenhouse gas changes in future projections.

The *lig127k* austral summer sea ice around Antarctica has a minimum in February in the MMM of $1.84 \pm 1.42$ mill $km^2$ (Fig. 4d). This is similar to the MMM of the *piControl* simulations (Fig. 4b). In both the *lig127k* and *piControl*, the models exhibit widely different sea ice areas (0.06 to 4.65 mill $km^2$) and distributions for the austral summer (Fig.

S5). Those models that simulate summer sea ice in the Weddell Sea in the *piControl* (Fig. S4) retain this sea ice in their *lig127k* simulation. The maximum austral winter sea ice around Antarctica also varies widely among the models, with the MIROC-ES2L simulating the smallest area (and seasonal cycle) and IPSLCM6 simulating the highest areal extent (and seasonal cycle) (Figs. 4b,d) in the *piControl* and *lig127k* simulations. ACCESS-ESM1-5 has the greatest sensitivity to the *lig127k* forcings (Fig. 4f).


The consensus from the *lig127k* sea ice distributions is a reduced minimum (August-September) summer sea ice extent (defined as 15% concentration) in the Arctic (Fig. 6) as compared to the *piControl* simulations (Fig. 3). It is interesting to compare the MMM simulated summer sea ice extents in the *lig127k* simulations to the observed sea ice extents for 2000-2018 (black lines in Fig. 6). More than half of the models simulate a retreat of the Arctic minimum

(August-September) ice edge at 127 ka, similar to the average of the last 2 decades. The pattern of February-March

Southern Ocean sea ice extent is broadly similar in the *lig127k* simulations to 2000-2018, though 4 models simulate a larger sea ice area.

### 3.4 Precipitation responses

The seasonal character of the insolation anomalies results in enhanced summer monsoonal precipitation in the *lig127k* ensemble (relative to the *piControl* ensemble) over northern Africa, extending into Saudi Arabia, India and southeast Asia, and northwestern Mexico/southwestern U.S (Fig. 8a). In contrast, summer monsoonal precipitation decreases over South America, southern Africa, and Australia. The spread among models is large, however, as shown by the across ensemble standard deviations (Figs. 8b,d) and percentage changes in area-averaged precipitation during

the monsoon season for seven different regional monsoon domains for the individual *lig127k* simulations (Fig. 16a). The models generally agree on the sign of the percentage changes in the area-averaged precipitation rate during the monsoon season for the monsoon regions, except for the East Asian, South Asian, and Australian-Maritime Continent monsoons where some models are simulating increased monsoonal precipitation whereas others are showing decreases.


Over the tropical Pacific Ocean, reduced DJF precipitation over the ITCZ is a robust feature across the ensemble of *lig127k* simulations (Fig. 8c). The models simulate a shift of the tropical Atlantic ITCZ northward in JJA and southward in DJF, though with significant differences among the models of the ensemble (Figs. 8a,b). Over the Indian Ocean, the ensemble mean indicates more precipitation in DJF over the entire basin and less in JJA,

particularly in the central and eastern basin, though again with large standard deviations (Fig. 8).

Figure 9 shows the ensemble-averaged *lig127k* change in monsoon-related rainfall rate and global monsoon domain. Increases in the summer rainfall rate and areal extent of the North Africa and East Asia monsoon are clear, and are robust across the multi-model ensemble. The spread across the multi-model ensemble is considerable, though, for

the North African (NAF) monsoon, with the percentage change in the areal extent varying from ~40-120% (Fig. 16b) and the percentage change in the total amount of water precipitated in each monsoon season varying from ~70-140% (Fig. 16c). The models are in closer agreement for the East Asian monsoon (EAS), with the percentage change in the areal extent varying from ~10-35% (Fig. 16b) and the percentage change in the total amount of water precipitated in each monsoon season varying from ~25-40% (Fig. 16c). The *lig127k* and *piControl* simulations

produce more muted changes for the other monsoon regions in the MMM, with regards to the regional monsoon-related rainfall rate and the monsoon domains (Fig. 9). Four models (AWI-ESM-1-1-LR, AWI-ESM-2-1-LR, MPI-ESM1-2-LR, NESM3) in the *lig127k* ensemble include interactive vegetation. Even then, these 4 models generally fall within the spread of the models with prescribed vegetation for the three metrics and 7 monsoon regions (Fig. 16).

**4 Data reconstructions**

## 4.1 Marine temperatures

The *lig127k* climate model simulations are assessed using two complementary compilations of sea surface temperature (SST) anomalies at 127 ka (Tables S3–S5, S7), which are both individually based on stratigraphically consistent chronologies (Capron et al., 2017; Hoffman et al., 2017).

The multi-archive high-latitude compilation by Capron et al. (2014, 2017) includes 42 sea surface annual/summer temperature records with a minimum temporal resolution of 2 kyr for latitudes above 40°N and 40°S, along with 5 ice core surface air temperature records. In contrast, the global marine compilation by Hoffman et al. (2017) includes 186 annual, summer and winter SST records from the Atlantic, Indian and Pacific oceans, with a minimum temporal resolution of 4 kyr on their published age models. Note that, in addition to the annual microfossil assemblage SST

records calculated for 41 sites as the average of the summer and winter records with a model- and observation-consistent correction for annual offsets (Hoffman et al., 2017), we also provide here for these specific sites the updated seasonal (summer and winter) SST estimates on the Hoffman et al. (2017) age models. SST from marine cores are reconstructed in both compilations from foraminiferal Mg/Ca ratios, alkenone unsaturation ratios or microfossil faunal assemblage transfer functions (Capron et al., 2014, 2017; Hoffman et al., 2017).

To derive the LIG marine chronologies, both compilations make use of the climate model-supported hypothesis that surface-water temperature changes in the sub-Antarctic zone of the Southern Ocean (respectively in the North Atlantic) occurred simultaneously with air temperature variations above Antarctica (respectively Greenland) (Capron et al., 2014; Hoffman et al., 2017). The compilation by Hoffman et al. (2017) then uses basin-synchronous LIG changes in the oxygen isotopic composition of benthic foraminifera, as observed in previous studies of benthic

foraminiferal isotope changes across glacial terminations (Lisiecki and Raymo, 2009) within the same ocean basins, to align intra-basin chronologies. However, a major difference is the underlying reference chronology used in both compilations: the Antarctic Ice Core Chronology 2012 (AICC2012) (Bazin et al., 2013; Veres et al., 2013) in the compilation by Capron et al. (2014, 2017), and a chronology based on millennial-scale variations observed in independently-dated Asian speleothem records (Speleo-Age) (Barker et al., 2011) in the compilation by Hoffman et

al. (2017). Note that the two reference chronologies diverge by about 1 ka at 127 ka (Capron et al., 2017).

The two compilations then follow quite similar Monte Carlo approaches to propagate temperature and chronological uncertainties. Indeed, both compilations generate 1000 realizations of the site-specific surface temperature records to integrate the uncertainty on the temperature reconstruction's method, and both produce 1000 possible chronologies to propagate the relative age uncertainty related to alignment of records. For a given site, the temperature at 127 ka is

the temperature value directly taken at 127 ka in the compilation by Hoffman et al. (2017), using dated temperature timeseries interpolated every 1 ka. In the compilation of Capron et al. (2014, 2017), the temperature at 127 ka is taken as the median temperature averaged over the 128-126 ka period. Finally, temperatures anomalies relative to preindustrial are calculated in both cases for marine sites using the HadISST dataset (Rayner et al., 2003), over the intervals 1870-1899 CE and 1870-1889 CE, in the compilations by Capron et al. (2017) and Hoffman et al. (2017),

respectively. For both compilations, the provided 2-sigma uncertainties integrate errors linked to relative dating and surface temperature reconstruction methods.

Nevertheless, because of: (1) the different reference chronologies used, (2) the different tie-points and associated relative age uncertainties defined to derive the chronology of each site, and (3) the different calculation methods (Bayesian statistics versus linear interpolation between tie-points) used in the Monte Carlo age model analysis of

each site (despite apparently relatively similar approaches), the two compilations by Capron et al. (2014, 2017) and by Hoffman et al. (2017) are listed as such in Tables S2-S5, S7. Implications of these methodological differences on the inferred 127 ka values are best illustrated when comparing the surface temperature timeseries deduced from the two different approaches for a same North Atlantic (62°N) site: at 127 ka, a temperature offset of ~2°C is observed between the two reconstructions (see Figure 4 of Capron et al., 2017).

**4.2 Ice core temperatures**

Surface air temperature records for one site (NEEM) on the Greenland ice sheet and four sites on the Antarctic ice sheet are deduced from ice core water isotopic profiles (Capron et al., 2014, 2017) (Tables S2 and S4). For ice cores, preindustrial conditions are estimated using borehole temperature measurements for Greenland, and 1870-1899 CE water isotopic profiles for Antarctica (Capron et al., 2017). Temperatures are again the median for the 126-128 ka

period, and are considered to represent annual averages. Uncertainty is estimated using the same Monte Carlo procedure as was used for the marine cores in the compilation of Capron et al. (2017). Because it uses the same reference timescale (AICC2012) the ice core dataset can be considered coherent with the marine SST dataset of Capron et al (2017).

**4.3 Terrestrial temperatures**

Calibrated, well-dated reconstructions of Last Interglacial temperatures over the continents are quite limited. We have assembled two distinct compilations of continental air temperature reconstructions: a dataset of air temperatures over Europe at 127 ka based on Brewer et al. (2008), and a compilation of peak Last Interglacial summer temperatures reconstructed at Arctic sites from pollen and insect assemblages (Table S6). For both we report anomalies comparing reconstructed temperatures with preindustrial climate estimated from 1871-1900.

In Europe, favorable geological conditions have led to the accumulation of numerous LIG sediment sequences from a variety of depositional environments (Tzedakis, 2007). These include former kettle lakes overlying late Saalian (MIS 6) till, depressions left by the penultimate alpine glaciation or local ice caps, and volcanic crater lakes or tectonic grabens mainly in the unglaciated south. Over several decades, a substantial body of pollen evidence has provided an insight into the LIG vegetational development across Europe. A number of pollen-based climate reconstructions on

reference sequences have been attempted, using a variety of methods. However, differences between underlying assumptions and data employed (e.g. taxon presence-absence *versus* abundance) mean that results have been difficult to compare.

Here, we include data from one study that has applied a multi-method approach to assess combined uncertainties of reconstruction and age models on a set of reference pollen records (Brewer et al., 2008). The reconstruction methods used are (i) partial least squares, (ii) weighted average partial least squares, (ii) generalized additive models, (iv) artificial neural network, (v) unweighted modern analogue technique, (vi) weighted modern analogue technique, and (vii) revised analogue method using response surfaces. Timescales for the pollen sequences were developed by transferring the marine chronology to land sequences for certain pollen stratigraphical evens on the basis of joint pollen and palaeoceanographic analyses in deep-sea sequences on the Portuguese Margin and Bay of Biscay (Shackleton et al., 2003; Sánchez Goñi et al., 2008). With particular reference to constraining the 127 ka timeslice, the pollen stratigraphical events used were the onset of the *Quercus* (128.8±1 ka) and *Carpinus* (124.77±1 ka) expansions (Brewer et al., 2008). For each site, chronological uncertainties were estimated at each sample by randomly sampling an age from the range around each control point, fitting a linearly interpolated age model and repeating this 1000 times (Brewer et al., 2008). Reconstructions were made at 500 yr intervals by randomly sampling within the chronological uncertainties and reconstruction errors for each method, resulting in 1400 estimates for each time t (Brewer et al., 2008). Here we present the mean value and standard deviation for mean annual temperature, mean temperature of the coldest month and mean temperature of the warmest month across all sites for 127 ka. Of the fifteen terrestrial sites used by Brewer et al. (2008), eight were excluded due to uncertainties over their chronostratigraphical or chronological assignments, or because they did not extend to 127 ka.

The Arctic dataset compiles the most stratigraphically complete, well-constrained in time, calibrated summer temperature reconstructions published from above 65°N latitude. We report the mean of the two warmest consecutive reconstructions at each site, utilizing the original published models and reconstructions. For sites where both insect- and pollen-based temperature reconstructions have been published, or where multiple models have been applied to the same proxy, we report here the average of those reconstructions. We report the original published model uncertainties (e.g., root mean square error of prediction for weighted averaging models), including the most conservative (largest) model uncertainties for sites where multiple proxies/models are applied. This differs from error reporting for the European dataset above. Importantly, the Arctic compilation also differs from the other paleotemperature datasets used here, in that it reports the warmest LIG conditions registered at each site rather than temperatures at 127 ka. This approach was necessitated by the coarse temporal resolutions and chronologies of the North American Arctic reconstructions, which come from stratigraphically discontinuous deposits dated by [14]C (non-finite [14]C ages) and in some cases luminescence or tephrochronology. In contrast to the North American Arctic sites, in northern Finland (Sokli) and northeast Russia (El'gygytgyn) correlative dating provides continuous chronologies. The reported peak warmth at those sites occurred at ~125 and 127-125 ka, respectively (Melles et al., 2012; Salonen et al., 2018). Reconstructed temperature at Sokli at 127 ka was ~1°C lower than the peak temperature reported here from that site. The Greenland ice core-derived temperature reconstruction from NEEM complements the Arctic terrestrial dataset, but it reflects annual rather than summer-specific climate.

485    Despite an abundance of LIG pollen records from Eurasia and various attempts at pollen-based climate
       reconstructions (e.g. compilations Velichko et al., 2007; Turney & Jones 2010), chronological and methodological
       uncertainties continue to complicate comparisons with climate model outputs. The lack of spatial coherence in the
       European temperature reconstructions may reflect depth-age model issues at individual sites, which implies that the
       127 ka timeslice had not been correctly identified.  An alternative approach would have been to select peak
temperatures from a wider interval (e.g. 127±2 ka) and assume that these are quasi-synchronous.  In addition, the
       Arctic reconstruction may be skewed towards warmer temperatures than the models, given that by definition this
       reconstruction is reporting the warmest period from each Arctic site, rather than the 127 ka timeslice.

## 4.4 Arctic sea ice

       A summary of LIG sea ice data obtained from marine cores in the Arctic, Nordic Seas and northern North
Atlantic, their interpretation, and comparison to the *lig127k* simulations can be found in Kageyama et al. (2020). The
       sea ice records are derived from dinoflagellate cysts, subpolar foraminers, and ostracods.

## 4.5 Precipitation

       Compilation of the existing proxy evidence for LIG precipitation have been presented for the Northern Asian and
       circum-Arctic region (CAPE, 2006; Kim et al., 2010; Velichko et al., 2008). Recently, a compilation with near-global
coverage was presented in Scussolini et al. (2019), including 138 proxy sites based on different types of proxies and
       archives, mostly from pollen, lacustrine sediment composition, speleothem, and multi-proxy reconstructions. This, in
       contrast to previous work, aimed to select proxy signals approximately corresponding to 127 ka, in order to facilitate
       comparison with results from the *lig127k* simulations. The main patterns that emerge, about precipitation change
       between the LIG and the pre-industrial/recent past, are near-ubiquitous higher LIG annual precipitation over the NH
(Fig. 13). Exception to this are individual sites in western North Africa, the Levant, northern South America, Borneo,
       the northwest of modern United States and Alaska, northern Scandinavia, and northern Siberia. Over the SH, the
       proxy signal is more irregular: Australia and the west coast of South America have proxies predominantly indicating
       higher precipitation in the LIG, sites in the rest of South America indicate lower precipitation or no change and over
       southern Africa changes are geographically more heterogeneous.

## 5    Model-data comparisons

### 5.1 Temperature

       Figures 10 to 12 compare the 127 ka temperature reconstructions discussed in Section 4 to the MMM and individual
       models. Details can be found in Tables S2-S7.

NH high-latitude terrestrial temperature proxies for the boreal summer (JJA) match the large warming in the *lig127k*
       MMM for most sites (Fig. 10a), except for Lake CF8 in Baffin Island and Wax Lips Lake in northeastern Greenland
       (Figure 12e, Table S6). These estimates are from subfossil midges and use published climatic and biogeographic
       calibration for calculating the mean temperature of the warmest month, rather than JJA, and represent the peak LIG

temperatures and not necessarily 127 ka. The only model that simulates the warming reconstructed for these two sites

is EC-Earth3-LR, though its *lig127k* simulation overestimates the warming farther south. Over Europe, the temperature proxies show generally positive anomalies for JJA, but these are often smaller than those of the *lig127k* MM (Fig. 10a). The *lig127k* MMM DJF temperatures over North America and Eurasia are significantly colder with respect to preindustrial, except over western Europe (Fig. 10c). The proxies for the latter show a mixed signal. The MMM indicates much warmer surface temperatures in DJF over the Arctic Ocean, Baffin Bay, and Labrador and

Greenland Seas, which cannot be evaluated given the available reconstructions (Fig. 10c). Annually, the MMM shows notable warming for Greenland and the ocean surrounding it (Fig. 10e). The range of warming is significant for site poleward of 50°N (Fig. 11a, Table S2). For the marine sites south of Greenland and near Iceland, the warming simulated by the individual models bracket the proxy estimate. For Greenland, all models are within the 2-sigma uncertainty for the NEEM ice core.


Over the North Atlantic, the MMM and proxy JJA temperature anomalies are generally in good agreement (Fig. 10a). The exceptions are in the northwestern North Atlantic and the Nordic Sea, where the Capron data suggest significant cooling. This mismatch could be associated with meltwater from potentially remnant ice sheets over Canada and Scandinavia, ice sheets that are not incorporated by the *lig127k* simulations. EC-Earth3-LR, HadGEM3-GC31-LL,

and ACCESS-ESM1-5 simulate the greatest warming at the 3 northernmost sites (poleward of 68N) in the Norwegian Sea, with EC-Earth3-LR warming outside the 2-sigma uncertainty range of the proxy JJA temperature anomalies (Fig. 12d, Table S5).

The marine reconstruction of Capron et al (2017) provides evidence of significant LIG warm temperature anomalies

for the austral summer (DJF) near New Zealand, which is neither exhibited by the *lig127k* MMM (Fig. 10d) nor the individual models which all cluster around little or no change in DJF temperature change (Fig. 12f, Table S7). This discrepancy suggests regional circulation changes not resolved by the models. The multi-model ensemble indicates austral winter (JJA) warming over the Southern Ocean and Antarctica, but the lack of proxies does not allow an assessment (Fig. 10b). The simulated annual temperature anomalies for the Antarctic ice cores are cooler than the

reconstructed values but generally fall within the 2-sigma uncertainties (Figs 10f, 12c, Table S4).

At lower latitudes (40°S – 40°N), marine proxy data from the Hoffman reconstruction are available (Fig. 11). They generally correspond with the MMM changes. The SST proxies from the tropical Atlantic match the colder MMM *lig127k* SSTs in DJF (Fig. 11b). The reconstructed cooling there in JJA is not captured in the MMM, leading to a

failure to also capture the annual mean signal (Figs 11a, c). Proxy indications of much warmer SSTs in the upwelling regions off the west coasts of South Africa, North America, and South America are not simulated by the models (Fig. 11, 12b, Table S3). The resolution of CMIP6 models is generally not adequate to properly simulate these narrow coastal upwelling regions.

**5.2 Precipitation**

As shown in a comparison with a smaller ensemble of 127 ka simulations (Scussolini et al., 2019), precipitation proxies from the global compilation largely match the annual precipitation from the models included in the *lig127k* MMM (Fig. 13a), with the overall hit rate comparing matches between the sign of the anomaly in the models and in the proxies of 65% (Fig. 13b). The agreement between the MMM and NH proxies is even higher over North Africa-Middle East (hit rate of 76%), North America-Greenland (hit rate of 78%), and South Asia (hit rate of 73%). It should be noted that the range across the individual model is quite large for North America-Greenland (hit rates of 45 to 90%) and South Asia (hit rates of 40 to 87%). Proxies and MMM weakly disagree over much of Europe, Central Asia and the region between them, where proxies indicate wetter LIG conditions or no change, and the MMM indicates somewhat drier conditions or no change (Fig. 13a). The overall MMM hit rate for Europe (68%) is much improved as compared to the smaller ensemble analyzed by Scussolini et al. (2019), but the range across the models is quite large (36% to 77%). Other instances of more regional disagreements in the NH are over the southern side of Northern Africa, with drier proxies and wetter models, and over the Mississippi basin, with a wetter proxy site and somewhat drier MMM. However, the coastal proxy sites near the Bay of Bengal, which show strongly drier conditions, are near the region of strongly drier conditions over the Atlantic suggesting a northward shift in the ITCZ. The agreement between the MMM and SH proxies is noticeably less than for the NH, with hit rates less than 50% except for South America with a hit rate of 89% (Fig. 13b). In the SH, proxies and models mostly agree over South America, while they disagree over Australia and in several locations over southern Africa, where many proxies and the MMM indicate wetter and drier LIG, respectively (Fig. 13a). The hit rates for individual models show that some models perform significantly better over Australia (Fig. 13b).

## 6  Comparison of the model sensitivities to the insolation anomalies at 127 ka and 6 ka

The large-scale features and evaluation of the CMIP6/PMIP4 *midHolocene* simulations in comparison to data reconstructions and in the CMIP5/PMIP3 endeavor can be found in Brierley et al. (2020). In this section, we briefly explore differences in the responses of surface temperature, monsoon precipitation, and Arctic sea ice to the different magnitudes and seasonal character of the insolation anomalies at 127 ka versus 6 ka.

### 6.1 Orbital forcing

The orbit at 6 ka was characterized by a smaller eccentricity than at 127 ka, similar to 1850 CE (Fig. 14). Perihelion occurred near the boreal autumn equinox as compared to close to the boreal summer solstice at 127ka and near aphelion at 1850CE. NH summer insolation anomalies at 6ka, ~5-10% greater than at 1850 CE, are considerably less than at 127ka (Fig. 1 and Fig. 14). In addition, the positive insolation anomalies of greater than 10% in the Arctic occur in July-August at 6ka as compared to May-August at 127 ka.  At SH mid- and high latitudes, the anomalous insolation anomalies are shifted to boreal fall/austral spring. As such, the orbital forcing on climate is expected to be stronger at 127 ka than at 6 ka.

### 6.2 Surface temperature responses

Figure 15 compares the MMM changes and standard deviations of ensemble changes of surface air temperatures for *lig127k*

and *midHolocene* simulations. In the tropics and Southern Hemisphere, the JJA zonal-average temperature anomaly is positive ($\sim$ +0.5°C) for the *lig127k* ensemble but negative ($\sim$ -0.5°C) in the *midHolocene* ensemble. The maximum JJA surface temperature anomalies occur at ~40°-65°N for both time periods but are significantly greater at 127 ka (over 3°C at 127 ka as compared to ~1°C at 6 ka). The DJF zonal-average surface temperature anomalies are near zero or negative south of 65°N for both time periods. Cryosphere and ocean feedbacks provide the memory for

positive surface temperature anomalies in DJF, even with negative insolation anomalies, with DJF Arctic surface temperatures averaging about 0.5°C higher in the *midHolocene* MMM and up to 3°C higher in the *lig127k* MMM than the *piControl*.

### 6.3 Precipitation responses

The signs of the percentage changes in the areal extents of the regional monsoon domain (Fig. 16b) and the percentage

changes in the total amount of water precipitated in each monsoon season (Fig. 16c) are similar for the *lig127k* and *midHolocene* simulations as compared to *piControl* simulations, but the responses are generally enhanced in the *lig127k* simulations as compared to the *midHolocene* simulations. Both time periods show greater areal extent and total amount of water precipitated for the NAF and EAS monsoons, with the *lig127k* MMMs outside the *midHolocene* quartile range. Similarly, the Australian-maritime Continent, South Africa, and South America

monsoons show greater reductions of areal extent and total water precipitated in the *lig127k* simulations than in the *midHolocene* simulations as compared to the *piControl* simulations. Both time periods show a mixed simulated response of the North American monsoon (NAMS).

### 6.4 Arctic sea ice responses

The boreal insolation anomalies at 6 ka enhance the seasonal cycle of Arctic sea ice, though much less so than in the

*lig127k* simulations (Fig. 17). None of the models currently in this analysis have the Arctic becoming ice-free in their *midHolocene* simulations. Similar to the analyses of the ensemble of *lig127k* simulations (see Section 3.3, Brierley et al. (2020)) also found that in the ensemble of CMIP6 *midHolocene* simulations, the summer sea ice reduction in the Arctic is correlated to the magnitude of annual warming over the Arctic but has little Arctic-wide relationship with the simulated preindustrial sea ice extents.

## 7  Discussion

The Tier 1 *lig127k* experiment was designed to address the climate responses to stronger orbital forcing (relative to the *midHolocene* experiment) using the same state-of-the-art models and following a common experimental protocol. At 127 ka, atmospheric greenhouse gas levels were similar to those of the preindustrial period, land ice likely only remained over Greenland and Antarctica, and the continental configurations were almost identical to modern.  In

addition, within uncertainties in chronology and dating, this time period allows data reconstructions for comparison to the model simulations allowing an assessment of responses to the large insolation changes. The 17 CMIP6 models that have completed the *lig127k* experiment are presented.

The CMIP6-PMIP4 *lig127k* simulations show warming and cooling over the continents during JJA and DJF, respectively, in response to the seasonal character of the insolation anomalies. The JJA MMM warming is greater than 6°C at mid-latitudes in North America and Eurasia, though with across ensemble standard deviations in excess of 2°C over the eastern U.S. and central Europe. The simulations exhibit a ~50% reduction and shift of Arctic minimum summer sea ice area to 3.1 mill km$^2$ in September for *lig127k*, though with a large range of 0.22 to 7.47 mill km$^2$. Positive temperature anomalies in the *lig127k* simulations annually in the Arctic and over the Southern Ocean, though with across ensemble standard deviations in excess of 2°C. The large spread across the models in their simulations of Arctic sea ice, even now with all models adopting a common experimental protocol, points to the need to better diagnose the atmosphere and ocean feedbacks that differ across the *lig127k* ensemble (Kageyama et al., 2020). As expected from the larger insolation anomalies in the *lig127k* than *midHolocene* simulations, the boreal summer responses in NH surface temperature and Arctic sea ice are amplified.

The CMIP6-PMIP4 *lig127k* simulations produce enhanced summer monsoonal precipitation and areal extent over northern Africa, which extends into Saudi Arabia, India and southeast Asia, and northwestern Mexico/southwestern U.S. In contrast, summer monsoonal precipitation decreases over South America, southern Africa, and Australia. The spread across the multi-model ensemble is particularly large for the North African monsoon, with the percentage change in its areal extent ranging from less than 50% to more than 150% and total amount of water precipitated during the monsoon season ranging from ~65% to more than 200%. The 4 models with interactive vegetation fall within the spread of the models with prescribed vegetation for the three metrics and 7 monsoon regions. The *lig127k* individual monsoon changes are of a similar sign, but a greater magnitude, to those seen in the *midHolocene* simulations (Brierley et al., 2020).

New syntheses for surface temperature and precipitation, targeted for 127 ka, have been developed for comparison to the CMIP6-PMIP4 *lig127k* simulations. The surface temperature reconstructions include two complimentary compilations of SST based on stratigraphically consistent chronologies, surface air temperatures from the Greenland and Antarctic ice sheets deduced from the ice core water isotopic profiles, continental air temperatures for Europe based on pollen records, and peak LIG summer temperatures in the Arctic inferred from pollen and insect assemblages. Anomalies were consistently computed comparing the reconstructed temperatures with observational-based preindustrial climate estimate from the end of the 19[th] century. A new precipitation reconstruction has expanded from previous regional compilations to now near-global coverage.

Over Canada, Scandinavia and parts of mid-latitude Europe, and the North Atlantic, the proxy and *lig127k* positive JJA temperature anomalies are in good agreement. The exceptions are in the northwestern North Atlantic and the Nordic Sea, where the Capron reconstruction (Capron et al., 2017) suggest significant cooling. The Capron reconstruction also provides evidence of significant positive DJF temperature anomalies over the Southern Ocean, which is not exhibited by the ensemble mean. These mismatches could be associated with regional ocean circulation

changes not resolved by the models as well as meltwater from potential remnant ice sheets over Canada and
Scandinavia as well as memory in the ocean of the H11 event (Govin et al., 2012; Marino et al., 2015), which the
*lig127k* simulations do not incorporate. The latter would lead to cooling in the North Atlantic and warming in the
Southern Ocean (Stone et al., 2016; Holloway et al., 2018).

The simulated annual temperature anomalies for the Greenland and Antarctic ice cores are cooler than the
reconstructed values but generally fall within the 2 sigma uncertainties. The *lig127k* Tier 1 experiment protocol
prescribed modern Greenland and Antarctic ice sheets rather than allowing them to evolve to smaller and lower ice
sheets of the *lig127k* climate. A modeling study with the HadCM3 (a CMIP3 model) demonstrated that the distinctive
peak in $\delta^{18}O$ observed in Antarctic ice cores at 128 ka was likely due to the loss of winter sea ice in the Atlantic,
Indian, and Pacific sectors of the Southern Ocean. To achieve this winter sea ice extent required forcing by the H11
meltwater event (Holloway et al., 2017, 2018). The CMIP6-PMIP4 Tier 2 LIG experiments (*lig127k-H11, lig127-
gris, lig127k*-ais) will allow modeling groups to explore the effects of the H11 meltwater event and the Antarctic and
Greenland ice sheets at their minimum LIG extent and lower elevations (Otto-Bliesner et al., 2017).

Other reasons for mismatches between the models and the reconstructions for temperature and precipitation will also
be explored with CMIP6-PMIP4 Tier 2 LIG experiments.  The CMIP6-PMIP4 Tier 2 *lig127k-veg* experiments will
consider the sensitivity of the responses to prescribed boreal forests in the Arctic and shrub/savanna over the Sahara,
separately and together. Incorporating these vegetation changes has been shown to impact the albedo and
evapotransporation on the surface energy and water budgets, reducing model and data mismatches at high latitudes
(Swann et al., 2010) and for the North African monsoon (Pausata et al., 2016). Recent results show large changes in
hydrology, with e.g. the possible existence of an extensive river network across Sahel and Sahara, therefore also
pointing to the need to prescribe or model vegetation changes to capture regional feedbacks (Scussolini et al., 2020).
Additionally, the CMIP6 models do not currently simulate changes to soil texture or color for different climate states.
A previous modeling study suggests that soil feedback can drive the African monsoon northward during interglacials
(Levis et al., 2004).

Temperature reconstructions are not available for many regions where the *lig127k* multi-model ensemble show
interesting responses to the *lig127k* forcing. These include the polar regions during their respective winter seasons:
Arctic and North Atlantic Oceans in DJF and Southern Ocean and Antarctica in JJA. Development of terrestrial
reconstructions for most continents and marine reconstructions for the Indian and Pacific Oceans would be useful for
assessing the model responses.

The CMIP6-PMIP4 *lig127k* experiment has potential implications for confidence in future projections. More than
half of the models simulate a retreat of the Arctic minimum (August-September) ice edge in their *lig127k* simulations
that is similar to the average of the last 2 decades (Fig. 6). Equilibrium Climate Sensitivity (ECS) (Table 2) and
simulation of August-September *lig127k* minimum Arctic sea ice area across the models show a significant (at the 0.5

level) correlation of -0.62 (Fig. 18). INM-CM4-8 with the smallest ECS of 1.8°C simulates the largest August-September *lig127k* Arctic sea ice area. CESM2 has a high ECS of 5.2°C (Gettelman et al., 2019); HadGEM3 similarly has a high ECS of 5.6°C (Guarino et al., 2020). Both predict an almost ice-free or ice-free Arctic in their *lig127k* experiments. Their predicted years of disappearance of September sea ice in the SSP8-8.5 scenario is 2038 and 2035, respectively (Guarino et al., 2020). Across CMIP6 models, Kageyama et al. (2020) noted a nearly linear relationship between the simulations of Arctic summer sea ice in their *1pctCO2* simulations at time of doubling and their *lig127k* simulations. With very limited Arctic sea ice proxies for 127 ka, and with evolving interpretation of the relationships of these proxies with sea ice coverage (Stein et al., 2017; Kageyama et al., 2020), it is currently difficult to rule out the high or low values of ECS from the proxy data.

Radiative perturbations on the Arctic system, even though related to summer insolation during the LIG and MH rather than greenhouse gas radiative forcing, might provide useful insights on the state of the future Arctic system (Schmidt et al., 2014). Using CMIP5 MH and RCP4.5 simulations from 10 climate models, Yoshimori and Suzuki (2019) examined the relevance of Arctic warming in the MH to that in the future. The radiative forcing in the RCP4.5 experiment is dominated by the elevated atmospheric $CO_2$ concentrations and is relatively uniform globally and seasonally. The radiative forcing in the MH associated with orbital forcing is seasonal, peaking in July-August. Yet for MH and RCP4.5, the largest Arctic warming and sea ice reduction occurs in late summer and early autumn. The surface energy balance analysis identifies local Arctic feedbacks associated with positive albedo feedback in summer and consequent increase in heat release from the ocean to atmosphere in autumn to be important contributors for both climate states.

Large differences exist among the models in the magnitude of the seasonal and annual surface temperature responses in the polar regions reflecting differences in the feedback processes represented by each model. These should be investigated. Warmer summer temperatures over Greenland, warmer oceans year-round surrounding Greenland, and reduced Arctic summer sea ice all have the potential to force a retreat of the ice sheet in the future. The *lig127k* results can be used to force Greenland ice sheet models, both one-way as included in the ISMIP6 protocols (Nowicki et al., 2016) and fully coupled to a climate model as is now being done by several modeling groups. With the availability of LIG ice and marine core records, LIG simulations with an evolving Greenland ice sheet will allow an assessment of the corresponding future projection simulations.

*Data availability.* The model outputs for the *lig127k* and *piControl* simulations are archived on the CMIP6 ESGF websites for all models included in this study, except for AWI-ESM-2-1-LR, HadGEM3-GC31-LL, and MPI-ESM1-2-LR. The model outputs for these three models are available on request from the modeling groups until it is posted on the ESGF. Table S1 lists the DOIs for the ESGF datasets. The data are included as tables in the Supplementary Material.

*Author contributions*. BLO-B, AZ, ECB and CB performed the bulk of the writing and analysis. YA–EW contributed data, text and analysis to the research. AA-O–WZ contributed the modeling simulations for the manuscript.

*Competing interests.* The authors declare no competing interests.

*Acknowledgements.* BLO-B, ECB and RT acknowledge the CESM project, which is supported primarily by the National Science Foundation (NSF). This material is based upon work supported by the National Center for Atmospheric Research (NCAR), which is a major facility sponsored by the NSF under Cooperative Agreement No. 1852977. Computing and data storage resources, including the Cheyenne supercomputer (doi:10.5065/D6RX99HX), were provided by the Computational and Information Systems Laboratory (CISL) at NCAR. CMB acknowledges the financial support of the Natural Environment Research Council through grant NE/S009736/1. AZ and CMB would like to thank Rachel Eyles for her sterling work curating the local replica of the PMIP archive at UCL.

CJRW acknowledges the financial support of the UK Natural Environment Research Council-funded SWEET project (Super-Warm Early Eocene Temperatures), research grant NE/P01903X/1 and the financial support of the Belmont-funded PACMEDY (PAlaeo-Constraints on Monsoon Evolution and Dynamics) project. AG acknowledges the support of the French national program LEFE/INSU (CircLIG project) and of the Belmont-funded ACCEDE project (ANR-18-BELM-0001-06). EW has received funding from the European Research Council under the Horizon 2020 research and innovation programme (grant agreement No 742224, WACSWAIN). This material reflects only the author's views and the Commission is not liable for any use that may be made of the information contained therein. EW is also funded by a Royal Society Professorship. P.S. acknowledges funding from the NWO (Nederlandse Organisatie voor Wetenschappelijk Onderzoek) under grant ALWOP.164. E.C. acknowledges financial support from the ChronoClimate project, funded by the Carlsberg Foundation. PB and MK acknowledge the HPC resources of TGCC allocated to the IPSL CMIP6 project by GENCI (Grand Equipement National de Calcul Intensif) under the allocations 2016-A0030107732, 2017-R0040110492, and 2018-R0040110492 (project gencmip6). This work was undertaken in the framework of the LABEX L-IPSL and the IPSL Climate Graduate School, under the "Investissements d'avenir" program with the reference ANR-11-IDEX-0004-17-EURE-0006. This study benefited from the ESPRI (Ensemble de Services Pour la Recherche à l'IPSL) computing and data center (https://mesocentre.ipsl.fr), which is supported by CNRS, Sorbonne Université, École Polytechnique, and CNES and through national and international, including the EU-FP7 Infrastructure project IS-ENES2 (grants. n° 312979). MS is funded by a scholarship from CEA and "Convention des Services Climatiques" from IPSL (https://cse.ipsl.fr/). https://cse.ipsl.fr/). KHN and CC acknowledges computational support provided by UNINETT Sigma2 - the National Infrastructure for High Performance Computing and Data Storage in Norway (projects nn4659k and nn9635k).

LM acknowledges support from the Australian Research Council FT180100606. The ACCESS-ESM 1.5 experiments were performed on Raijin at the NCI National Facility at the Australian National University, through awards under the National Computational Merit Allocation Scheme, the Intersect allocation scheme, and the UNSW HPC at NCI Scheme. QZ acknowledges the support from the Swedish Research Council (Vetenskapsrådet, grant no. 2013-06476 and 2017-04232). The EC-Earth simulations are performed on ECMWF's computing and archive facilities and on resources provided by the Swedish National Infrastructure for Computing (SNIC) at the National Supercomputer Centre (NSC) partially funded by the Swedish Research Council through grant agreement no. 2016-07213. Weipeng ZHENG acknowledges the financial support from National Key R&D Program for Developing Basic Sciences (Grant No. 2016YFC1401401), the Strategic Priority Research Program of Chinese Academy of Sciences (Grant Nos. XDA19060102 and XDB42000000) and the National Natural Science Foundation of China (Grants Nos. 91958201and 41376002), and the technical support from the National Key Scientific and Technological Infrastructure project "Earth System Science Numerical Simulator Facility" (EarthLab). MVG and LS acknowledge the financial support of the NERC research grant NE/P013279/1. SRB and DSYM acknowledge Meteo-France / DSI for providing computing and data storage resources. XS and CS acknowledge computing and data storage resources provided by the DKRZ for generation of the AWI-ESM-1 / AWI-ESM-2 and MPI-ESM-1-2 simulations. The Max Planck Institute for Meteorology in Hamburg is acknowledged for development and provision of the MPI-ESM as well as the ECHAM6/JSBACH, that provides the atmosphere and land surface component of AWI-ESM. GL acknowledges funding via the Alfred Wegener Institute's research programme PACES2. CS acknowledges funding by the Helmholtz Climate Initiative REKLIM and the Alfred Wegener Institute's research programme PACES2. XS acknowledges financial support through the BMBF funded PACMEDY and PalMOD initiatives. AA-O and RO acknowledge the financial supports from Arctic Challenge for Sustainability (ArCS) Project (grant JPMXD1300000000), Arctic Challenge for Sustainability II (ArCS II) Project (Grant Number JPMXD1420318865), JSPS KAKENHI grant 17H06104 and MEXT KAKENHI grant 17H06323, and the support from JAMSTEC for use of the Earth Simulator supercomputer. PM was supported by the state assignment project 0148-2019-0009. EV was supported by RSF grant 20-17-00190.

The authors acknowledge QUIGS (Quaternary Interglacials), a working group of Past Global Changes (PAGES), which in turn received support from the US National Science Foundation, Swiss National Science Foundation, Swiss Academy of Sciences, and the Chinese Academy of Sciences. We are grateful to the World Climate Research Programme (WCRP), which, through its Working Group on Coupled Modelling, coordinated and promoted CMIP6. We thank the climate modeling groups for producing and making available their model output, the Earth System Grid Federation (ESGF) for archiving the data and providing access, and the multiple funding agencies who support CMIP6 and ESGF. The Paleoclimate Modelling Intercomparison Project is thanked for coordinating the lig127k protocol and making the model-model and model-data comparisons possible within CMIP6. PMIP is endorsed by WCRP and PAGES.

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

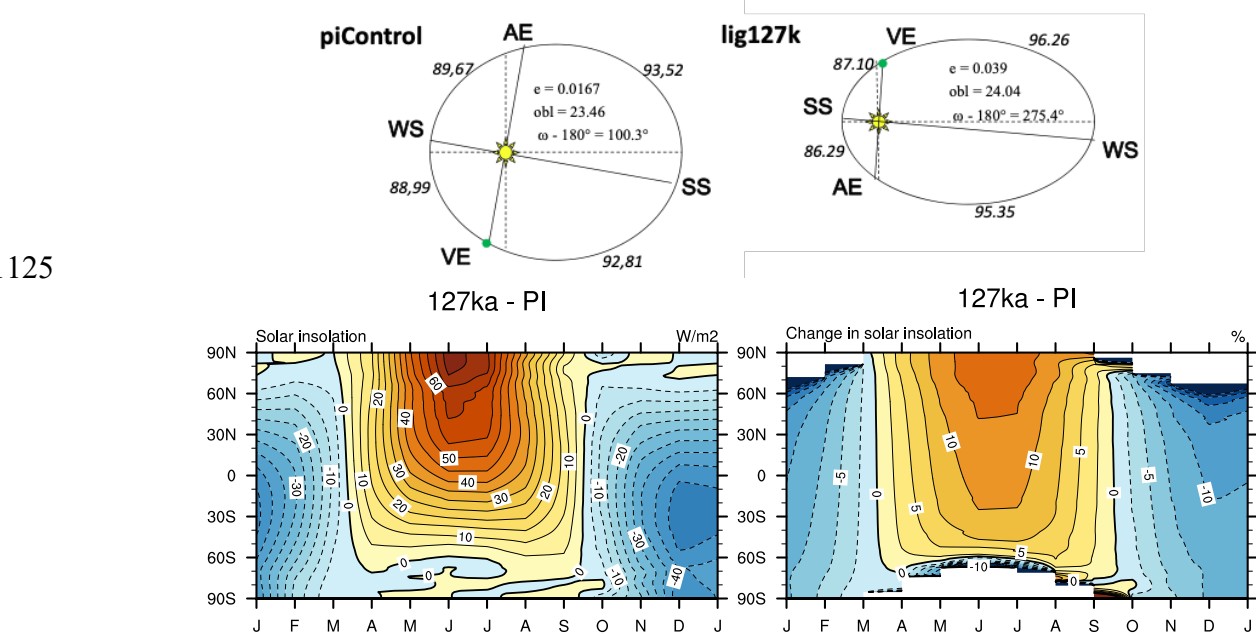

**Figure 1.** (top) Orbital configurations for the *piControl* and lig127k experiments. The number of days between the vernal equinox and summer solstice, summer solstice and autumnal equinox, etc are indicated along the periphery of the ellipse. (bottom) Latitude-month insolation anomalies 127 ka – 1850 in (left) W/m$^2$ and (right) percentage change from PI.



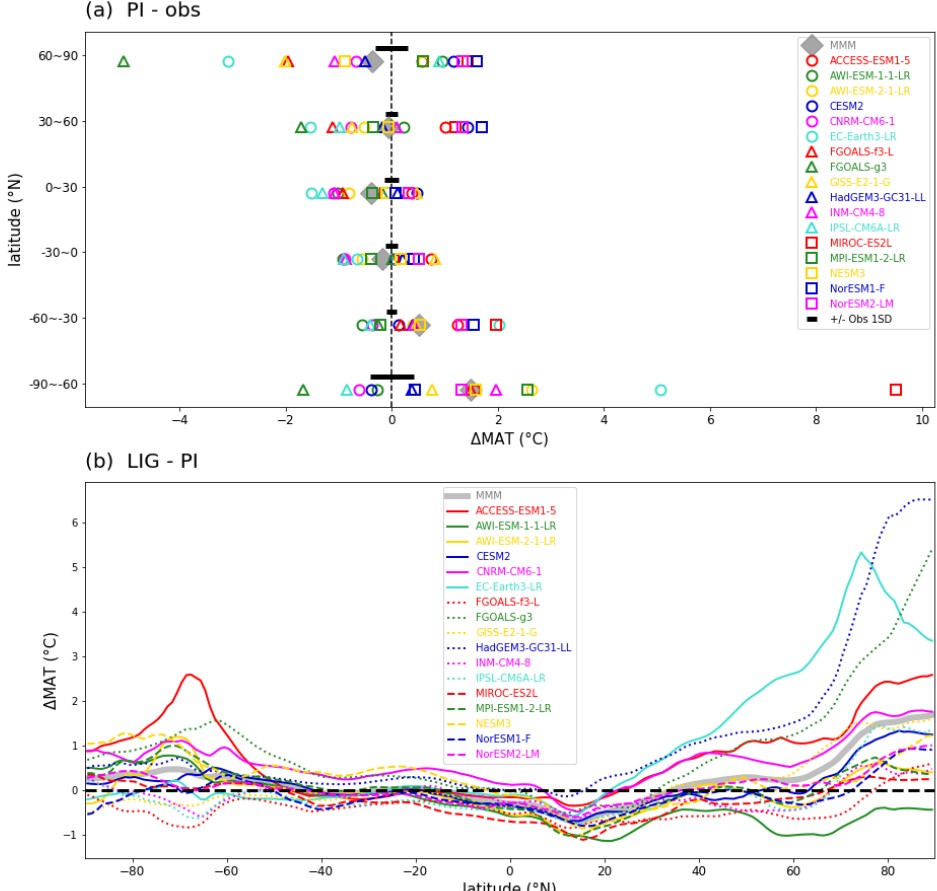

**Figure 2.** (a) Comparison of the preindustrial zonal mean temperature profile of individual climate models and MMM to the 1850-1900 observations. The area-averaged, annual mean surface air temperature for 30° latitude bands in the CMIP6 models and a spatially complete compilation of instrumental observations over 1850-1900 (black, Ilyas et al., 2017; Morice et al., 2012). (b) Changes in zonal average, mean annual surface air temperatures (*lig127k* minus *piControl)*.



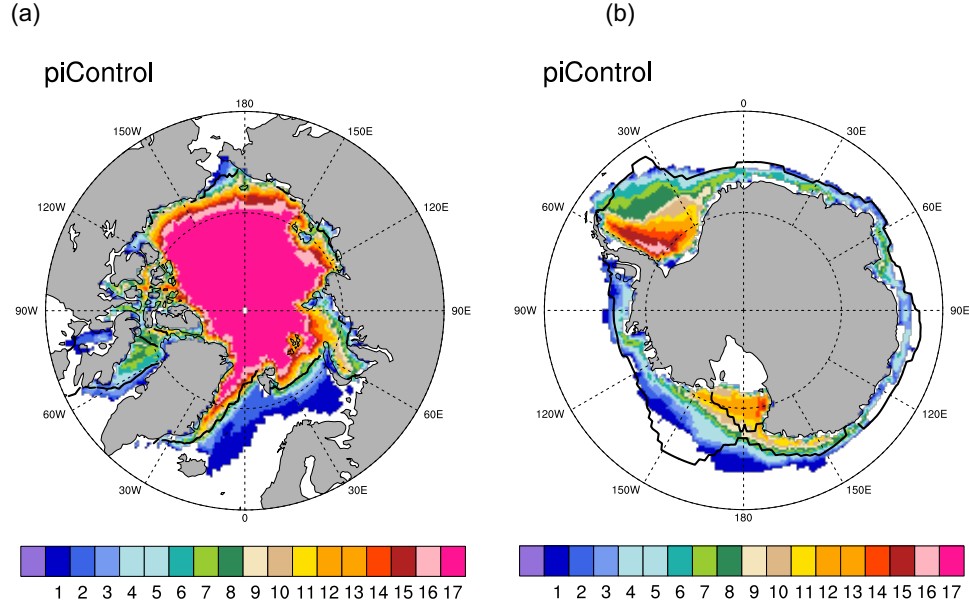

**Figure 3.** Comparison of the *piControl* sea ice distributions (a) in the Northern Hemisphere for August-September and (b) in the Southern Hemisphere for February-March. For each 1° x 1° longitude-latitude grid cell, the figure indicates the number of models that simulate at least 15% of the area covered by sea ice. The observed 15% concentration boundaries (black lines) are the 1870-1919 CE interval based on the Hadley Centre Sea Ice and Sea Surface Temperature (HadISST; Rayner et al., 2003) data set. See Figures S2 and S4 for individual model results.

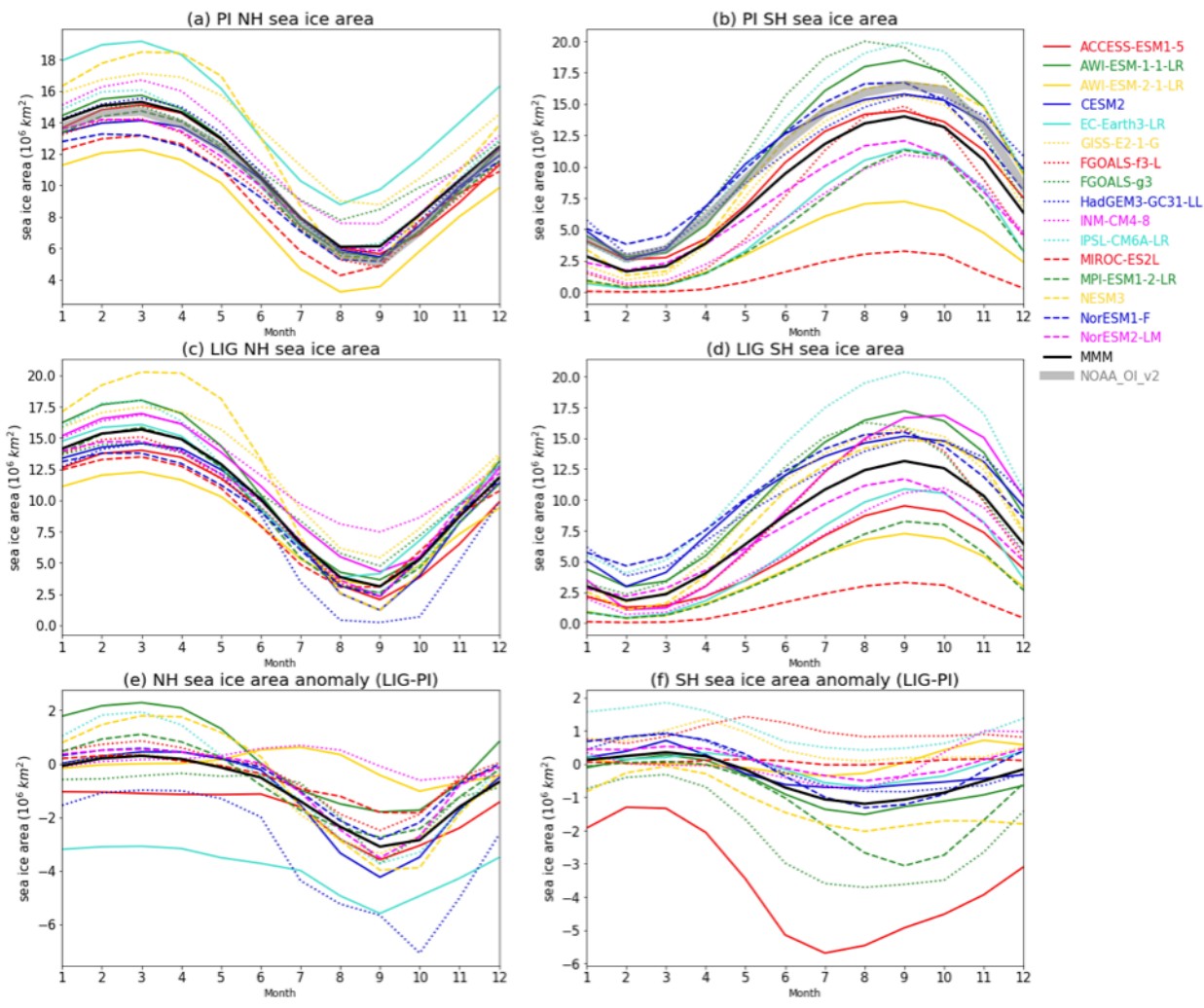


**Figure 4.** The simulated Arctic (left column) and Antarctic (right column) annual cycle of sea ice area of sea ice ($10^6$ km$^2$) for the (a,b) PI, (c,d) LIG, and (e,f) LIG minus PI. The monthly mean sea ice areas from the NOAA_OI_v2 data set for 1982-2001 (Reynolds et al., 2003) are shown in panels a,b.



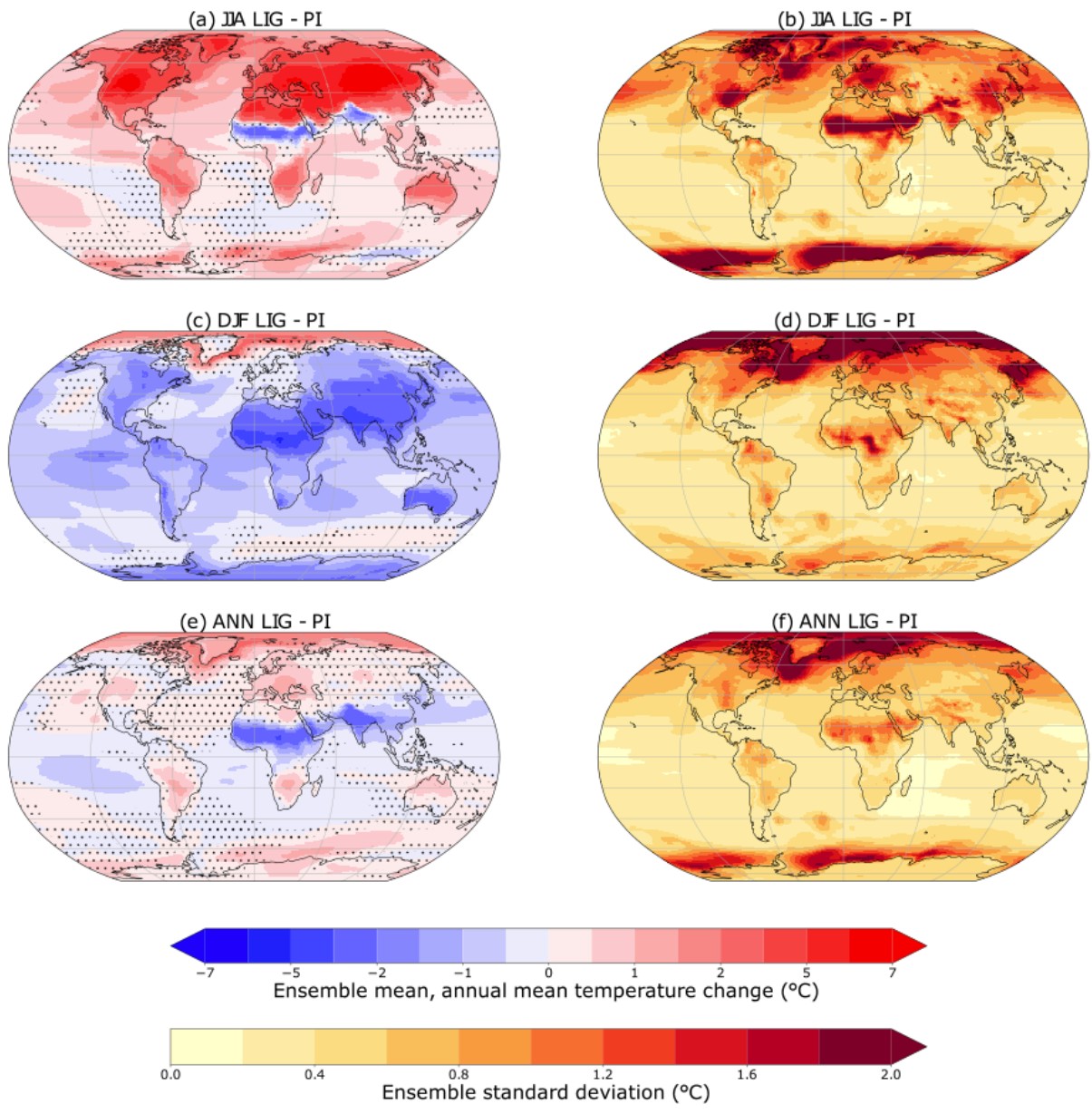

**Figure 5.** Multi-model ensemble average changes (left column) and across ensemble standard deviations (right column) of surface air temperatures (∘C) for *lig127k* minus *piControl*. Shown are June-July-August (a,b), December-January-February (c,d), and Annual mean (e,f) changes. Dots indicate where less than 12 (70%) of the 17 models agree on the sign of the change.




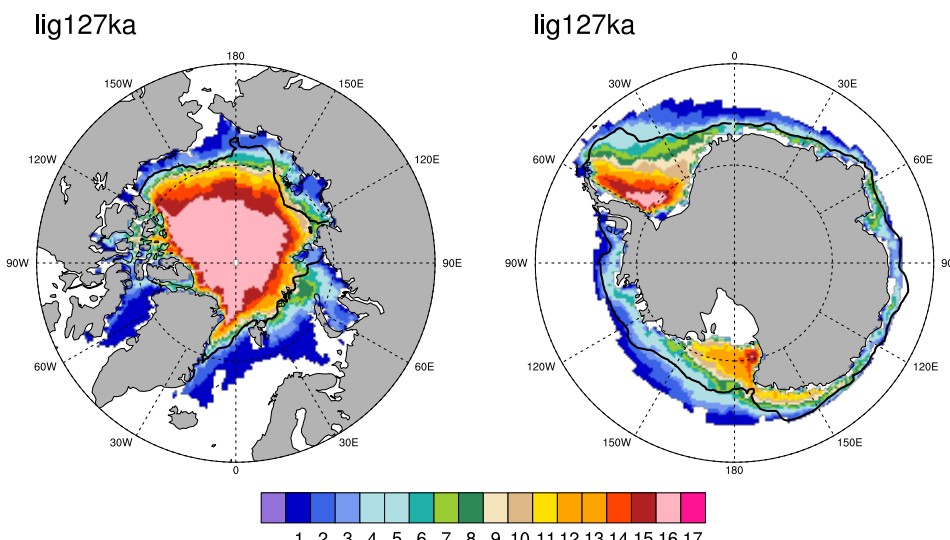

**Figure 6.** Comparison of the *lig127k* sea ice distributions (a) in the Northern Hemisphere for August-September and (b) in the Southern Hemisphere for February-March. For each 1° x 1° longitude-latitude grid cell, the figure indicates the number of models that simulate at least 15% of the area covered by sea ice. The average 15% concentration boundaries (black lines) are average for 2000-2018. See Figures S3 and S5 for individual model results.


## Northern Hemisphere August-September

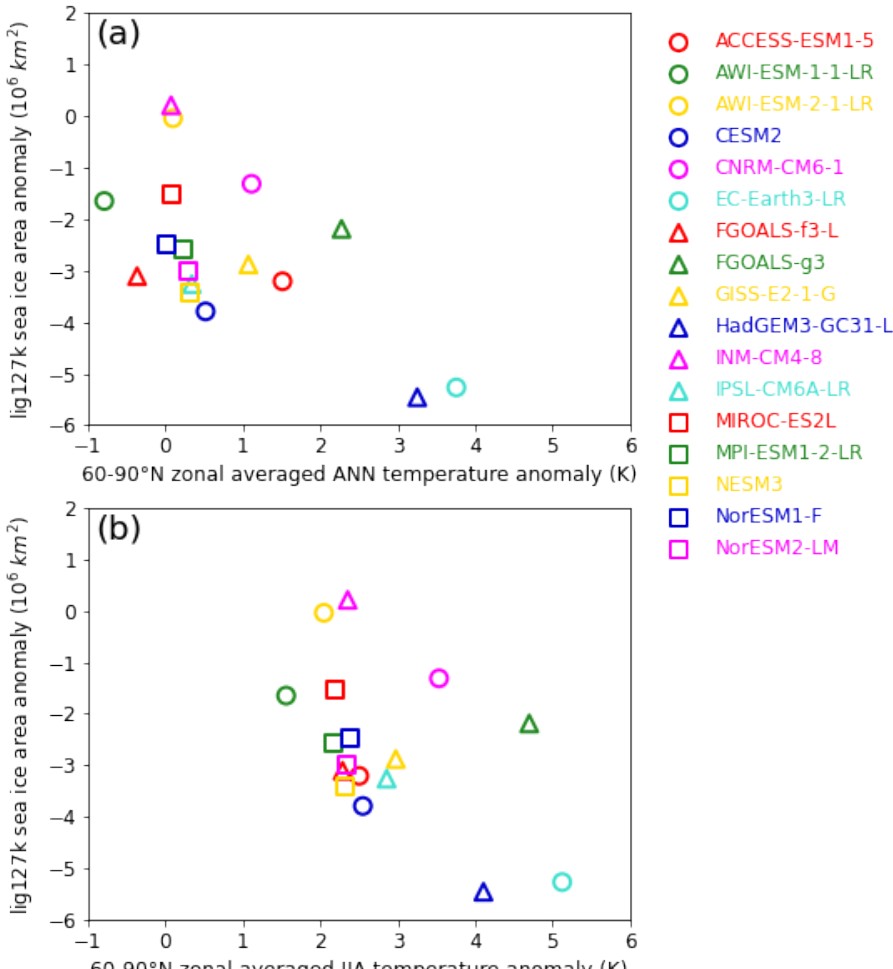

**Figure 7.** (a) lig127k August-September sea ice NH area anomaly ($10^6$ km$^2$) versus lig127k annual 60-90°N surface air temperature anomaly (°C); (b) lig127k August-September NH sea ice area anomaly ($10^6$ km$^2$) versus lig127k JJA 60-90°N surface air temperature anomaly (°C).





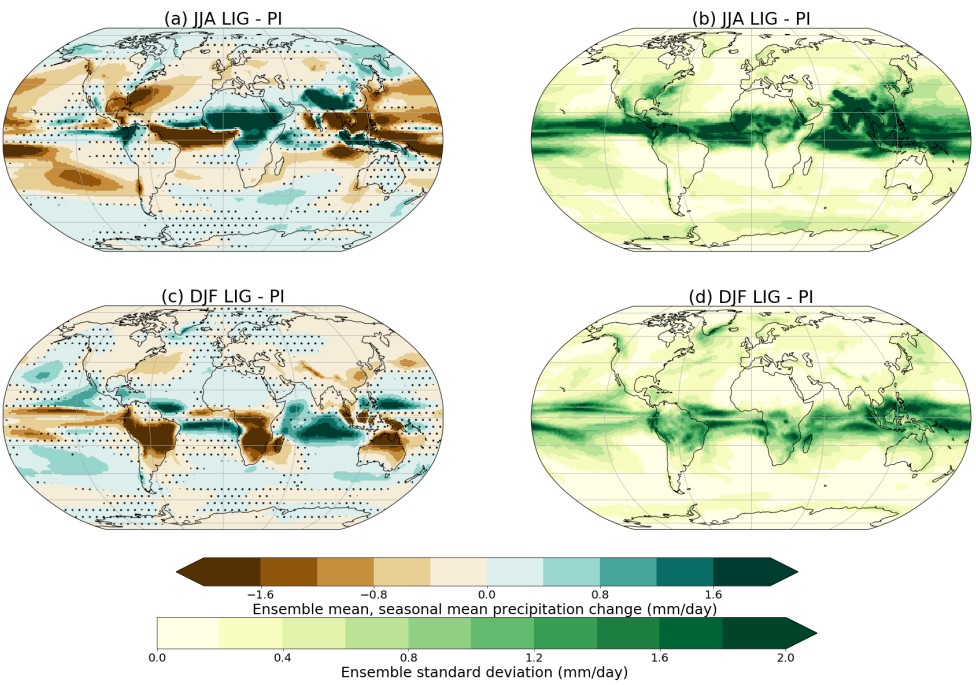

**Figure 8.** Multi-model ensemble average changes (left) and across ensemble standard deviations (right) of precipitation (mm/day) for *lig127k* minus *piControl.* Shown are June-July-August (a,b), December-January-February (c,d) changes. Dots indicate where less than 12 (70%) of the 17 models agree on the sign of the change.



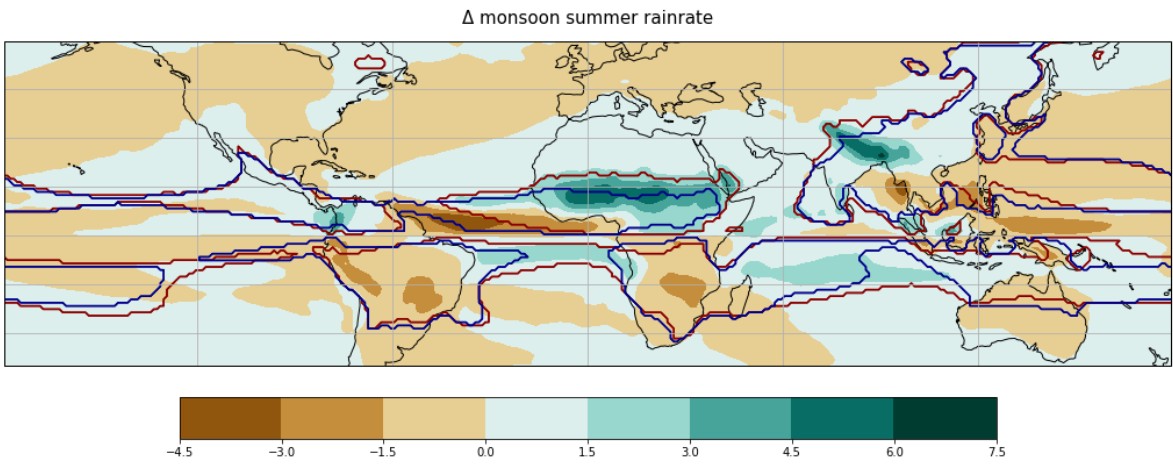

**Figure 9.** Ensemble-averaged Last Interglacial change in monsoon-related rainfall rate (in mm/day). Red and blue contours show the boundaries of *lig127k* and *piControl* monsoon domains, respectively, using the definitions of Wang et al. (2011).


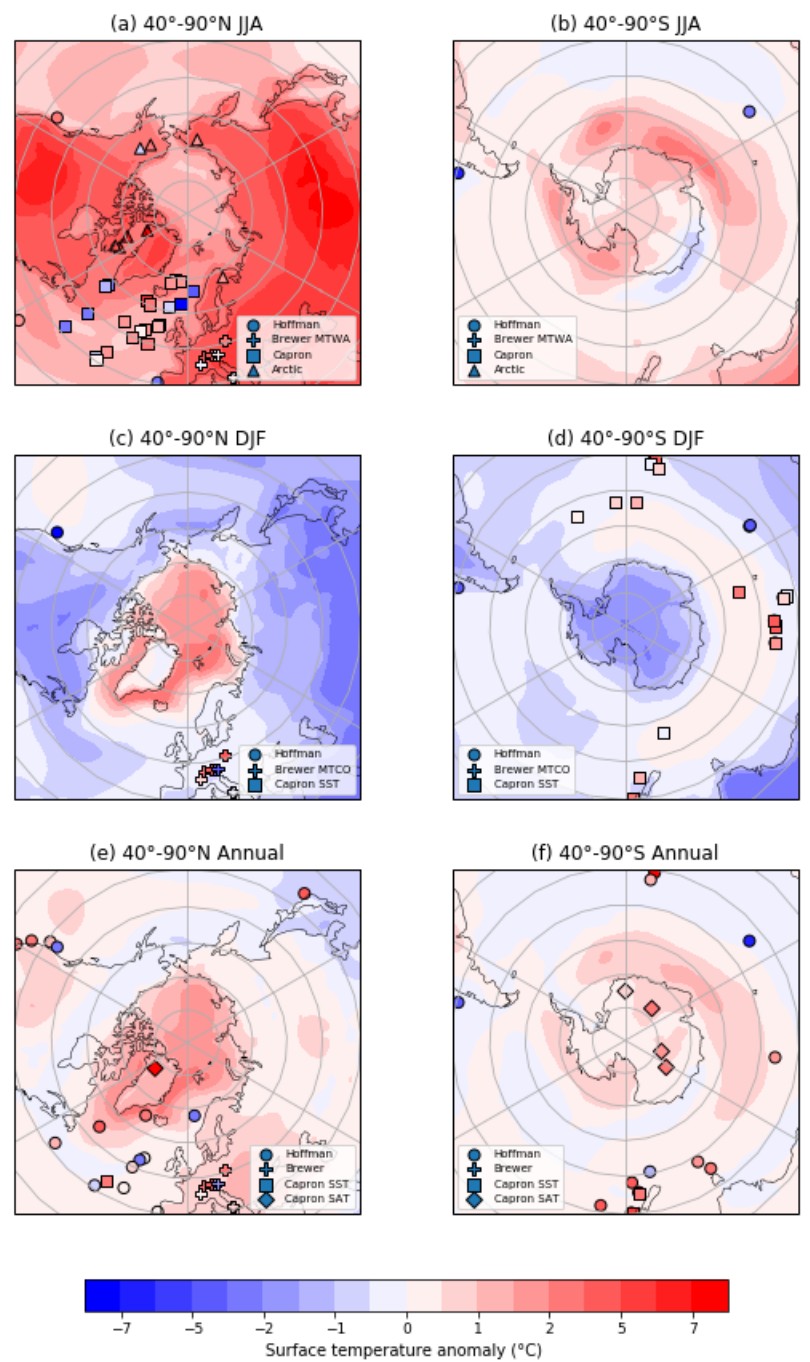

**Figure 10.** High-latitude surface temperature anomaly between 127 ka and Preindustrial from models (ensemble average in colors) and proxies (filled markers): circles for the compilation by Hoffman et al. (2017), squares and diamonds for marine sites and ice cores, respectively, of the compilation by Capron et al. (2014, 2017), pluses for the compilation of Brewer et al. (2008), and triangles for the Arctic compilation. (a,b) June-July-August, (c,d) December-January-February, (e,f) Annual. The preindustrial reference is 1850 CE for model anomalies and for the data is 1870-1899 for Capron, Brewer, and Arctic; 1870-1889 for Hoffman.

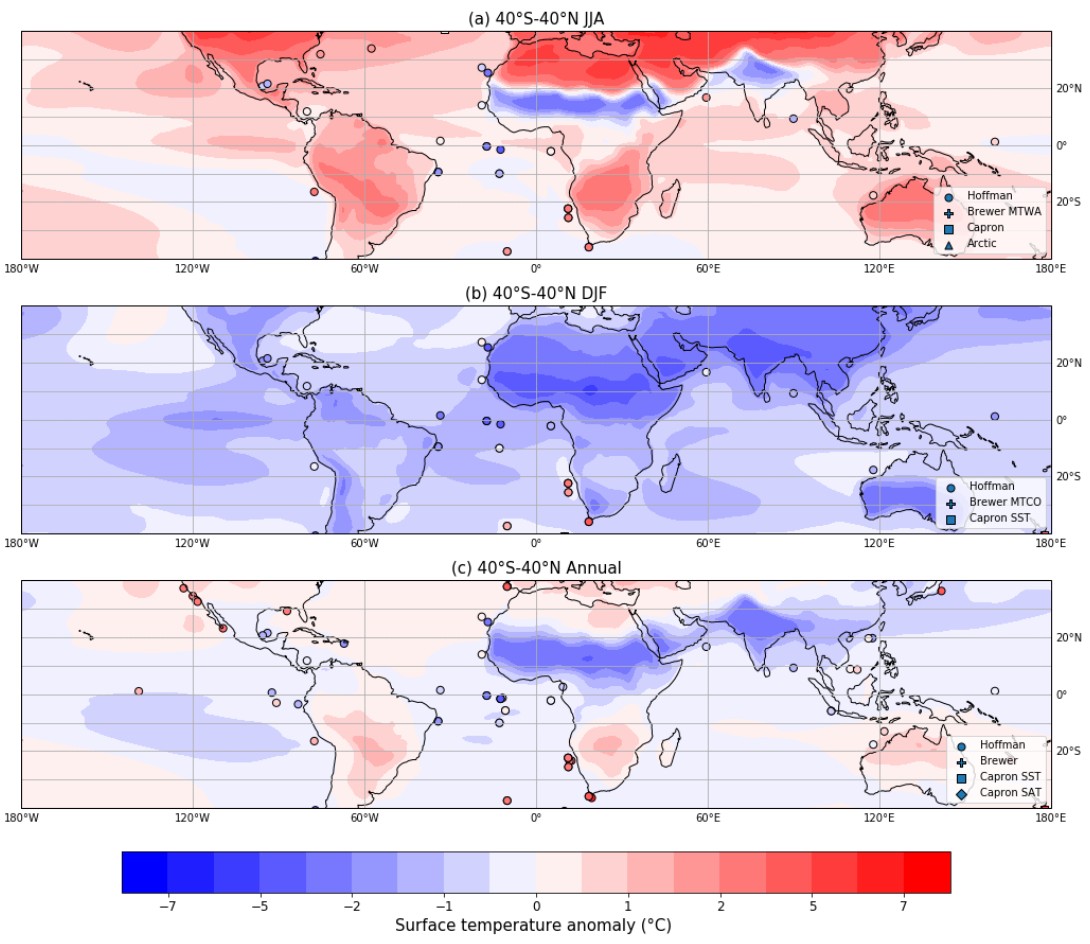


**Figure 11.** Same as Figure 10 but for low-latitude (40°S – 40°N) surface temperature.


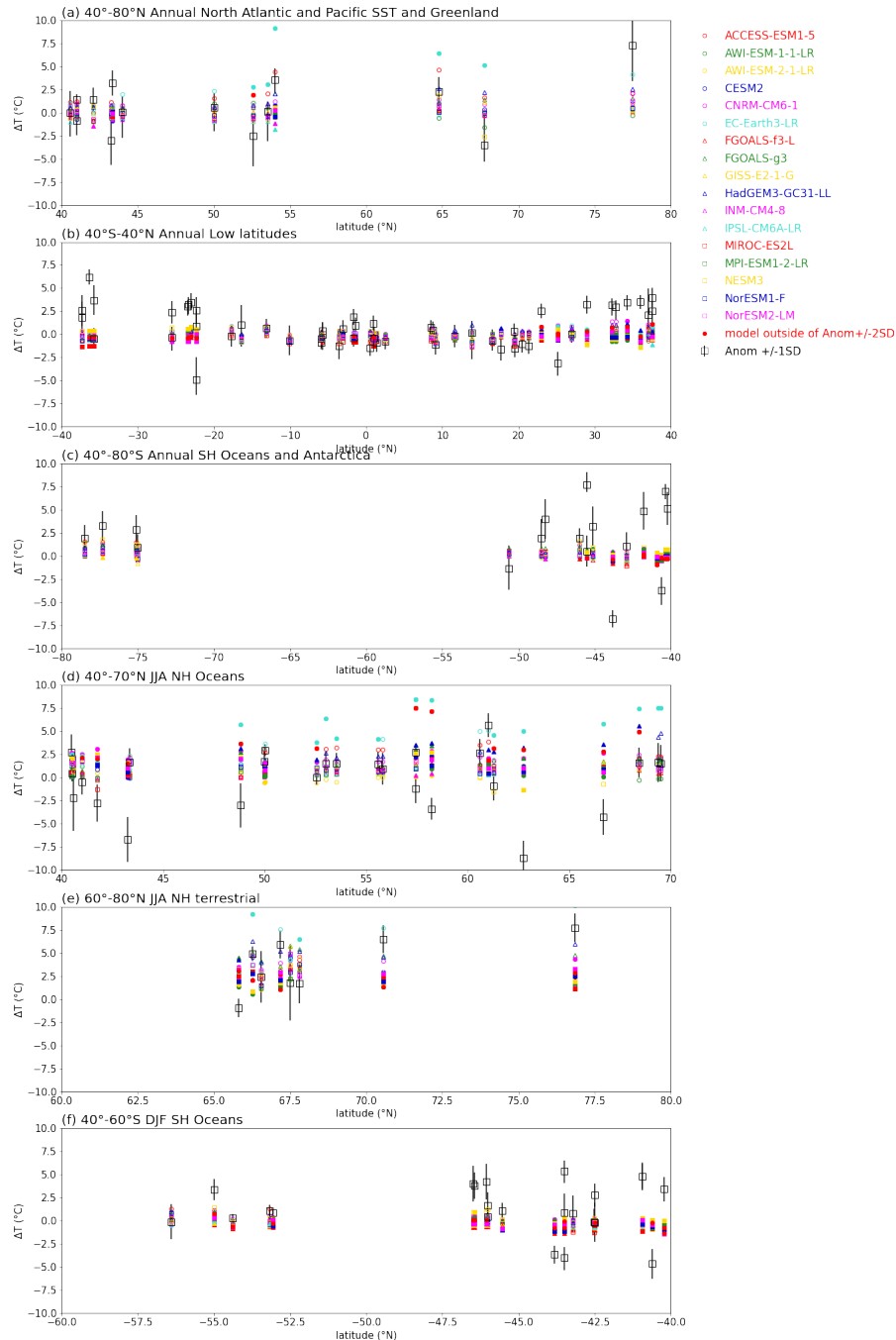

**Figure 12.** Comparison of proxy estimates of surface temperature anomalies (±1 standard deviation) with modelled temperature anomalies at the locations of the proxy data. Annual anomalies for (a) 40°-80°N, North Atlantic and Pacific SST and Greenland, (b) 40°S-40°N SST, (c) 40°-80°S SH Ocean SST and Antarctic SAT. Seasonal anomalies for (d) 40°-70°N, JJA NH Oceans, (e) 60°-80°N, JJA NH Terrestrial, and (f) 40°-60°S, DJF SH Oceans. All units are °C. Data and model values supporting this figure can be found in Supplementary Tables S2-S7.


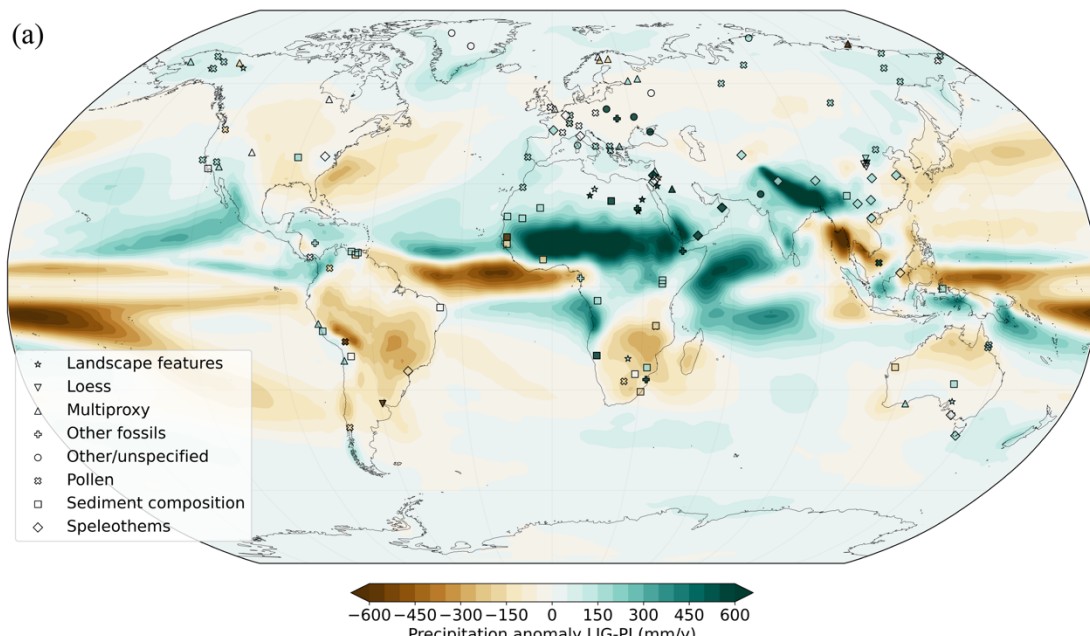

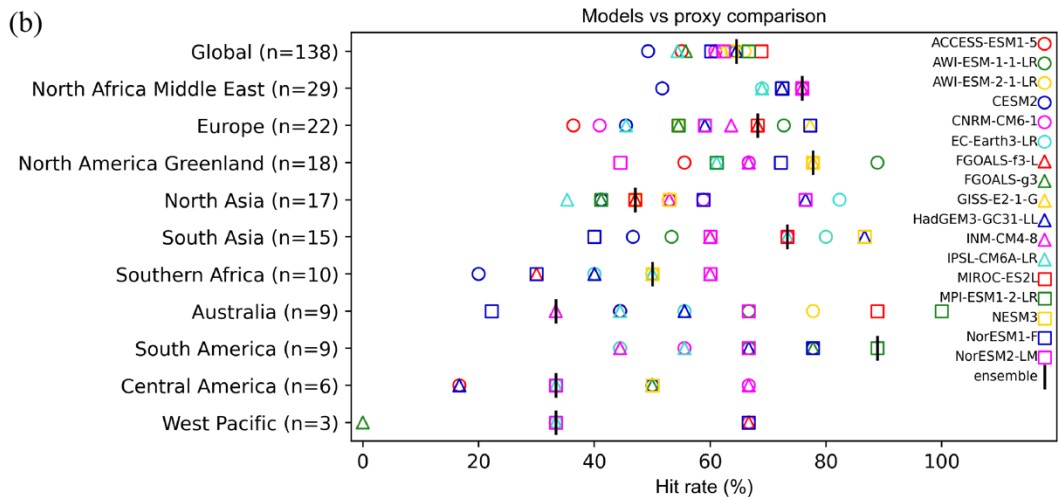

**Figure 13.** (a) Annual precipitation anomaly between 127 ka and Preindustrial, from model simulations (ensemble average in
contoured colors) and from proxies (filled markers). Green colors indicate higher precipitation during the 127ka from models or
proxies, and conversely for brown colors. Proxy anomalies are on a semi-quantitative scale: dark green (much wetter LIG), light
light green (wetter), white (no noticeable anomaly), light brown (drier), and dark brown (much drier). Different markers represent
different types of proxy records as specified in the legend. Proxy reconstruction from Scussolini et al. (2019). (b) Annual
precipitation anomaly between LIG and PI, comparison of models and proxies. The hit rate is the percentage of matches between
the sign of the anomaly in the models and in the proxies. n is the number of model-proxy comparisons per region.


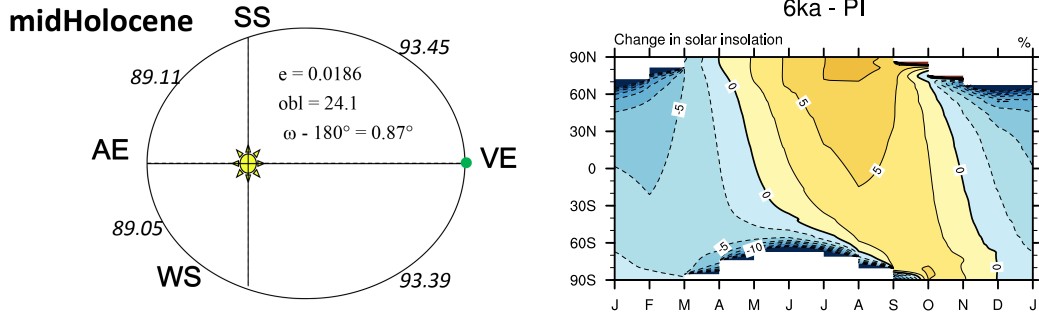


**Figure 14.** (left) Orbital configuration for the *midHolocene* (6ka) experiment. (right) Latitude-month insolation anomalies 6 ka–1850 as percentage change from PI.



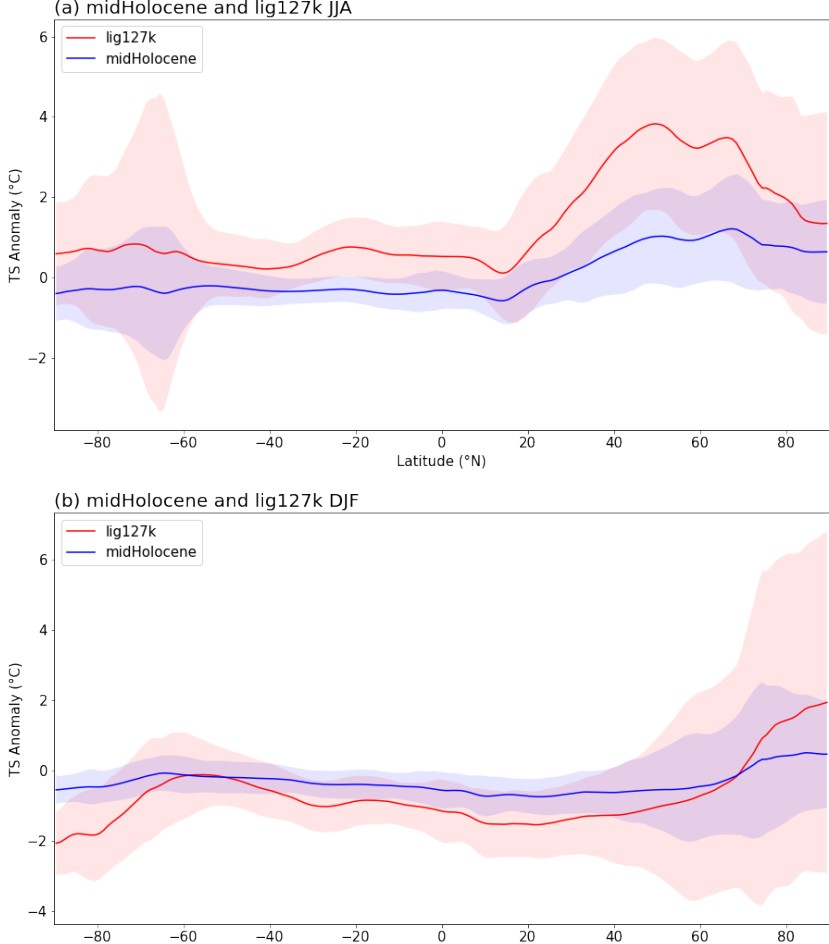


**Figure 15.** Multi-model ensemble mean and two standard deviation, zonal surface air temperature anomalies (°C) for
*midHolocene* and *lig127k* simulations for JJA and DJF (see Brierley et al., 2020 for more details on *midHolocene* simulations).
Note that 14 models completed both the *midHolocene* and *lig127k* experiments. Three models: ACCESS-ESM1-5, AWI-ESM-2-
1-LR, CNRM-CM6-1 completed only the *lig127k* experiment, while three models: MRI-ESM2-0, UofT-CCSM-4, BCC-CSM1-2
completed only the *mid-Holocene* experiment.

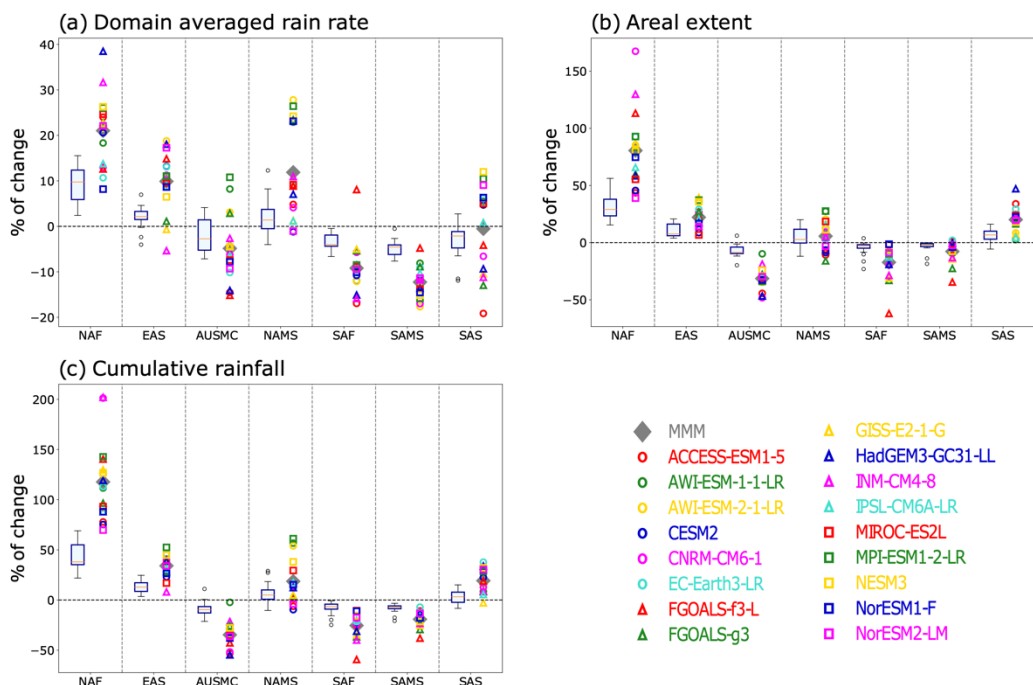

**Figure 16. Relative changes in MMM and individual *lig127k* monsoons.** Three different monsoon diagnostics as computed for each of seven different regional domains for the individual CMIP6 *lig127k* simulations. The comparable results from the *midHolocene* simulations are shown with boxes and whiskers (for details see Brierley et al., 2020). (a) the percentage changes in area-averaged precipitation rate during the monsoon season; (b) the percentage change in the areal extent of the regional monsoon domain; (c) the percentage change in the total amount of water precipitated in each monsoon season (computed as the precipitation rate multiplied by the areal extent). The abbreviations used to identify each regional domain are: North America Monsoon System (NAMS), North Africa (NAF), Southern Asia (SAS) and East Asia (EAS) in the Northern Hemisphere and South America Monsoon System (SAMS), South Africa (SAF) and Australian-Maritime Continent (AUSMC) in the Southern Hemisphere. Note that 14 models completed both the *midHolocene* and *lig127k* experiments. Three models: ACCESS-ESM1-5, AWI-ESM-2-1-LR, CNRM-CM6-1 completed only the *lig127k* experiment, while three models: MRI-ESM2-0, UofT-CCSM-4, BCC-CSM1-2 completed only the *mid-Holocene* experiment.

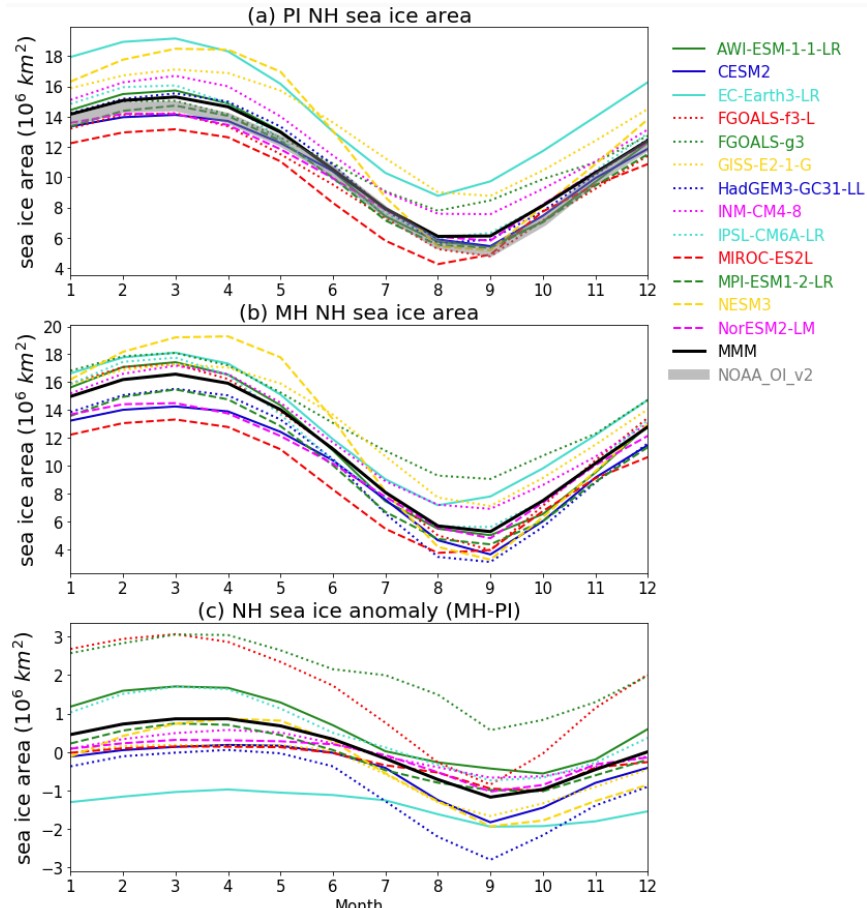

**Figure 17.** The Arctic annual cycle of area of sea ice greater than 15% ($10^6$ km$^2$) for the (a, top) PI, (b, bottom) MH for a subset of the models. Note that 14 models completed both the *midHolocene* and *lig127k* experiments. Three models: ACCESS-ESM1-5, AWI-ESM-2-1-LR, CNRM-CM6-1 completed only the *lig127k* experiment.

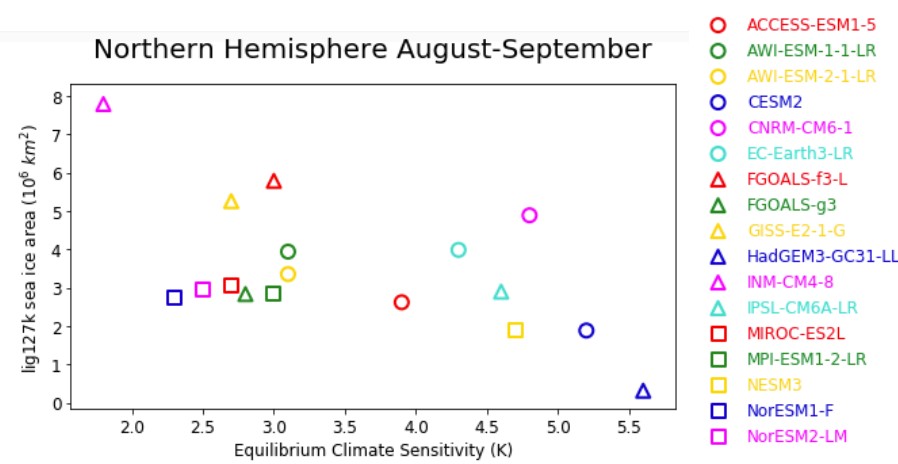

**Figure 18.** lig127k August-September NH sea ice area ($10^6$ km$^2$) versus Equilibrium Climate Sensitivity (ECS, K)

**Table 1.** Protocols: forcings and boundary conditions

|  | **1850 CE (DECK *piControl*)** | **127ka (*lig127k*)** |
|---|---|---|
| Orbital parameters[1] |  |  |
|    Eccentricity | 0.016764 | 0.039378 |
|    Obliquity (degrees) | 23.459 | 24.040 |
|    Perihelion - 180 | 100.33 | 275.41 |
|    Vernal equinox | Fixed to noon on March 21 | Fixed to noon on March 21 |
| Greenhouse gases |  |  |
|    Carbon dioxide (ppm) | 284.3 | 275 |
|    Methane (ppb) | 808.2 | 685 |
|    Nitrous oxide (ppb) | 273.0 | 255 |
|    Other GHG gases | CMIP DECK *piControl* | 0 |
| Solar constant (Wm$^{-2}$) | TSI: 1360.747 | Same as *piControl* |
| Paleogeography | Modern | Same as *piControl* |
| Ice sheets | Modern | Same as *piControl* |
| Vegetation | CMIP DECK *piControl* | Prescribed or interactive as in *piControl* |
| Aerosols Dust, Volcanic, etc. | CMIP DECK *piControl* | Prescribed or interactive as in *piControl* |

1350    [1]The term 'orbital parameters' is used to denote the variations in the Earth's eccentricity and longitude of perihelion as well as changes in its axial tilt (obliquity).

**Table 2.** Summary of CMIP6-PMIP4 models in this intercomparison.

| Climate Model | Institution Name | Citation for Model Description | Equilibrium (Effective) Climate Sensitivity[1] | Citation for lig127k Experiment and Notes[2] |
|---|---|---|---|---|
| ACCESS-ESM1-5 | UNSW and CSIRO | Ziehn et al., 2017 and Ziehn et al., 2020 | 3.9°C | Fixed vegetation with interactive LAI, Prescribed aerosols |
| AWI-ESM-1-1-LR | AWI | Sidorenko et al., 2015 | 3.1°C | Interactive vegetation |
| AWI-ESM-2-1-LR | AWI | Sidorenko et al., 2019 | 3.1°C | Interactive vegetation, prescribed aerosols |
| CESM2 | NCAR | Danabasoglu et al., 2020 | 5.2°C | Otto-Bliesner et al., 2020<br><br>Crops and Urban areas removed; Prescribed potential vegetation (crops and urban areas removed), interactive LAI, Simulated dust |
| CNRM-CM6-1 | CNRM-CERFACS | Voldoire et al., 2019<br>Decharme et al., 2019 | 4.8°C | PI atmospheric GHGs<br>Prescribed vegetation and aerosols |
| EC-Earth3-LR | Stockholm University | Doescher and the EC-Earth Consortium, in preparation, 2020 | 4.2°C | Zhang et al., 2020<br><br>Prescribed vegetation and aerosols |
| FGOALS-f3-L | CAS | He et al., 2020 | 3.0°C | Zheng et al., 2020<br><br>Prescribed vegetation and aerosols |
| FGOALS-g3 | CAS | Li et al., 2020 | 2.8°C | Zheng et al., 2020<br><br>Prescribed vegetation and aerosols |
| GISS-E2-1-G | NASA-GISS | Bauer and Tsigardis (2020) | 2.7°C | — |
| HadGEM3-GC31-LL | BAS | Kuhlbroat et al., 2018<br>Williams, et al., 2018. | 5.6°C | Guarino et al., 2020<br>Williams et al., 2020<br><br>Prescribed vegetation and aerosols |
| INM-CM4-8 | INM RAS | Volodin et al., 2018 | 1.8°C | Prescribed vegetation<br>Simulated dust and sea salt |
| IPSL-CM6A-LR | IPSL | Boucher, et al., 2020 | 4.6°C | Prescribed vegetation, interactive phenology, prescribed aerosols |
| MIROC-ES2L | AORI<br>University of Tokyo | Hajima et al., 2020 | 2.7°C | Ohgaito et al., 2020<br>O'ishi et al., 2020<br><br>Prescribed vegetation and aerosols |

| | | | | |
|---|---|---|---|---|
| MPI-ESM1-2-LR | AWI<br>MPI-Met | Giorgetta et al., 2013 | 3.0°C | Scussolini et al., 2019<br><br>Interactive vegetation<br>Prescribed aerosols |
| NESM3 | NUIST | Cao et al. (2018) | 4.7°C | Interactive vegetation<br>Prescribed aerosols |
| NorESM1-F | Norwegian Climate<br>Centre, NCC | Guo et al., GMD, 2019 | 2.3°C | Prescribed vegetation and aerosols |
| NorESM2-LM | Norwegian Climate<br>Centre, NCC | Seland et al., 2020 | 2.5°C | Prescribed vegetation and aerosols |

[1]ECS uses the Gregory method from a 150-year run of an instantaneously quadrupled $CO_2$ simulation (Meehl et al., 2020, Wyser et al., 2020)

[2]Unless otherwise noted, Prescribed vegetation and aerosols are as in each model's *piControl* simulation

1360

**Table 3. Metrics for surface air temperature change (°C ) for CMIP6-PMIP4 *lig127k* simulations**

1365

| Climate Model | Global | Global Land | Global Ocean | NH | NH Land | NH Ocean | SH | SH Land | SH Ocean | NH Meridional Gradient[1] | SH Meridional Gradient[2] |
|---|---|---|---|---|---|---|---|---|---|---|---|
| ACCESS-ESM1-5 | 0.33 | 0.42 | 0.29 | 0.43 | 0.34 | 0.48 | 0.23 | 0.58 | -0.05 | 1.61 | 1.89 |
| AWI-ESM-1-1-LR | -0.25 | -0.47 | -0.16 | -0.55 | -0.81 | -0.37 | 0.04 | 0.25 | -0.08 | 0.38 | 0.86 |
| AWI-ESM-2-1-LR | -0.20 | -0.34 | -0.14 | -0.39 | -0.59 | -0.25 | -0.01 | 0.20 | -0.14 | 0.8 | 0.78 |
| CESM2 | -0.11 | -0.16 | -0.09 | -0.22 | -0.31 | -0.16 | 0.00 | 0.18 | -0.08 | 1.02 | 0.47 |
| CNRM-CM6-1 | 0.4 | 0.39 | 0.4 | 0.33 | 0.15 | 0.46 | 0.46 | 0.89 | 0.26 | 1.21 | 0.55 |
| EC-Earth3-LR | 0.45 | 0.71 | 0.34 | 0.99 | 0.92 | 1.03 | -0.07 | 0.32 | -0.17 | 3.94 | 0 |
| FGOALS-f3-L | -0.48 | -0.57 | -0.44 | -0.60 | -0.77 | -0.48 | -0.37 | -0.16 | -0.35 | 0.3 | -0.28 |
| FGOALS-g3 | 0.38 | 0.6 | 0.29 | 0.38 | 0.51 | 0.29 | 0.48 | 0.89 | 0.24 | 2.42 | 1.14 |
| GISS-E2-1-G | -0.12 | -0.1 | -0.13 | -0.07 | -0.17 | 0.00 | -0.18 | 0.06 | -0.20 | 1.59 | -0.11 |
| HadGEM3-GC31-LL | 0.56 | 0.71 | 0.49 | 0.89 | 0.76 | 0.97 | 0.22 | 0.62 | 0.08 | 3.08 | 0.37 |
| INM-CM4-8 | -0.2 | -0.3 | -0.15 | -0.30 | -0.54 | -0.14 | -0.09 | 0.20 | -0.12 | 0.45 | -0.23 |
| IPSL-CM6A-LR | -0.29 | -0.3 | -0.29 | -0.29 | -0.43 | -0.19 | -0.30 | -0.03 | -0.31 | 0.89 | -0.02 |
| MIROC-ES2L | -0.4 | -0.55 | -0.33 | -0.52 | -0.73 | -0.38 | -0.26 | -0.12 | -0.29 | 0.92 | 0.55 |
| MPI-ESM1-2-LR | -0.12 | -0.24 | -0.07 | -0.33 | -0.54 | -0.19 | 0.10 | 0.42 | -0.05 | 0.95 | 0.83 |
| NESM3 | 0.07 | -0.02 | 0.11 | -0.25 | -0.43 | -0.12 | 0.39 | 0.86 | 0.22 | 0.83 | 0.57 |
| NorESM1-F | -0.24 | -0.35 | -0.2 | -0.33 | -0.55 | -0.18 | -0.15 | 0.08 | -0.21 | 0.59 | 0.24 |
| NorESM2-LM | -0.11 | -0.04 | -0.14 | -0.13 | -0.13 | -0.12 | -0.09 | 0.16 | -0.16 | 0.69 | 0.39 |
| | | | | | | | | | | | |
| **Mean** | **-0.02** | **-0.04** | **-0.01** | **-0.06** | **-0.20** | **0.04** | **0.02** | **0.32** | **-0.08** | **1.27** | **0.47** |
| **Std dev** | **0.32** | **0.44** | **0.28** | **0.48** | **0.54** | **0.45** | **0.26** | **0.34** | **0.19** | **1.00** | **0.55** |
| **Max** | **0.56** | **0.71** | **0.49** | **0.99** | **0.92** | **1.03** | **0.48** | **0.89** | **0.26** | **3.94** | **1.89** |
| **Min** | **-0.48** | **-0.57** | **-0.44** | **-0.60** | **-0.81** | **-0.48** | **-0.37** | **-0.16** | **-0.35** | **0.30** | **-0.28** |

[1] 60-90N minus 0-30N
[2] 60-90S minus 0-30S