# Peer review of "Large-scale features of Last Interglacial climate: Results from evaluating the *lig127k* simulations for CMIP6-PMIP4"

_Climate of the Past, 2019_

## Referee Comment (RC1) · Anonymous Referee #1 · 27 Jan 2020

The manuscript covers a topic that can be of great value to the climate research community and the authors present a wide variety of data sets and analyses that can yield valuable insights. However, I feel that at present the manuscript is too incomplete for a thorough review and in the following I will confine myself to a number of overall remarks that I think the authors should carefully address in a resubmission of the manuscript.

-> Ensure that the set of climate simulations is complete and that whenever possible the same set is used in all figures and analyses. When this is not the case it should be clearly mentioned. How was the set of 127k simulations determined? Why are for instance some models from the recent publication by Scussolini et al. not included?

[Figure]

-> A similar remark for the 6k simulations: ensure that whenever possible the same set is used in all figures and analyses (in figure 17 there are only 5 6k models, is that all of them?).

-> Are the comparisons of 127k and 6k based only on models that performed both experiments?

-> If an analysis can only be done on a subset of models, make it plausible that the results are not impacted too much by leaving out a substantial number of models.

-> Show the robustness of your results and statements by statistical measures. Are the multi-model-mean results significantly different from zero considering the inter-model spread? Are the presented correlations significant?

-> Indeed PI biases in models can be important, but if these data are presented then a much clearer link to the ensuing 127k and 6k results should be made. Can some of the palaeo results be explained by the PI biases? Are the biases consistent across these different experiments? Present more real analyses like in figure 8, not only a description of the results.

-> Hardly any underlying mechanisms are described in the manuscript. Perhaps this is outside of the scope of this manuscript, but it appears to me that many of the mechanisms that can explain the PI, 6k and 127k biases or results have already been discussed elsewhere and references should thus be provided.

-> Why is no comparison of the 127k results made with previous last interglacial experiments? For instance Lunt et al. (2012)? Perhaps these experiments did not exactly target 127k, but this could well be of second-order importance for the results of the experiments. Did the model response change from PMIP3 to PMIP4?

-> Clarify what this manuscript adds to previous more targeted manuscripts on for instance precipitation (Scussolini et al, 2019) or sea ice (Kageyama et al., 2019).

---

## Referee Comment (RC2) · Anonymous Referee #2 · 11 Mar 2020

Summary

The authors provide an overview of the lig127k simulations prepared for CMIP6 and PMIP4. They show the main global features of an ensemble of 17 coupled climate models, including the temperature, precipitation and sea-ice response. The article will provide an important reference for the CMIP process and a starting point for further more detailed lig127k analysis. Having said that, the present text could in my opinion better convey what has been learned about the Last Interglacial climate and/or about these climate models. Therefore, I recommend publication once these messages have been brought out, either through more detailed discussion or with further analyses.

[Figure]

Main comments

Several questions feel unanswered at the end of this manuscript, e.g. why might models be getting the incorrect pattern of change in some regions (e.g. is it really all down to missing freshwater or dynamic vegetation)? What factors might contribute to the relatively large spread in simulated responses? How robust are the seasonal versus annual reconstructions and which one of these tells us most about the model deficiencies? Why does the sea-ice loss scale with ECS? How do models with interactive vegetation or LAI differ from the fixed vegetation models?

Whilst I realise that a full treatment of each of these questions could generate a paper by itself, more discussion would be extremely valuable, especially given that the author list brings together a list of experts for each model and in the palaeoclimate archives.

Other comments

If I understand correctly, you are showing that models with higher ECS also show stronger Arctic sea-ice loss in LIG simulations. This despite lower GHG levels in the lig127k simulations. I think this is interesting, but requires more analysis.

It feels incomplete to omit HadGEM3 from the sea-ice ECS comparison, especially as this model has the largest polar warming in the ensemble. Please can you include this in the analysis?

Lines 595-604: I don't really agree with the main point here. None of the dynamic vegetation models in PMIP2 or CMIP5 showed an adequate precipitation response in North Africa, so why should this differ for the last interglacial?

Table 2: The details of the models included needs to be completed before publication. What is the HadGEM3 ECS and how were vegetation and aerosols treated in FGOALS, GISS-E2-1-G and NESM3 and CNRM?

Fig. 11-13: I think some assessment of the model uncertainty and paleo reconstruction uncertainty is required. It's not clear from these figures whether the multi-model mean

is good but the individual models are biased etc.

Technical corrections

Line 550: "though with significant differences among the models". Line 552: "but with substantial differences among the ensemble" Line 555: "though again with a large spread across 555 the model ensemble" Line 562: "The spread across the multi-model ensemble is particularly large for the North African monsoon" Line 585: "However, the model spread is large"

Please quantify these.

Line 583: "The most consistent picture from the temperature proxies representing annual conditions is warmer LIG temperatures over Greenland and Antarctica" What do you mean here? Consistent between model simulations and reconstructions, or consistent within the reconstructions?

Line 627: "There appears to be a clear relationship between the ECS of each model and its simulation of August-September lig127k minimum Arctic sea ice extent"

I'm not sure this is accurate. The comparison in the first panel of figure 8 (which needs to be properly labelled) is moderate at best, but perhaps I have misunderstood the plots as the labels are inconsistent with the caption. The r2 would be useful here.

Figures

Fig. 3: a) I can see why you have offset the models and observations in the latitudinal direction, but this could lead to confusion. Would this plot not work better as anomalies? Mostly what we see here is that the models capture the latitudinal temperature gradient. Additionally what is the uncertainty on the observations?

b) Please can you join the circles in the lower plot with lines, so that we can see the integrated latitudinal change in each model separately. For example, this might show that a model is the warmest at high-latitudes but is in the middle of the ensemble

elsewhere?

Fig. 8: the y-axes are incorrectly labelled in all but the first panel. There are also grey lines between some of the panels.

Fig. 16: ACCESS-ESM appears to show close to no change in these figure panels - can you double check this?

———————————————

---

## Author Comment (AC1) · 18 May 2020

We would like to thank you for your comments about our manuscript. We believe that the revisions we plan to implement should satisfactorily address your comments.

The first main set of comments in this review relates to the set of simulations included in the figures and analysis and included several queries.

1. Why not same set of simulations used in all figures and analyses? The submission of the paper was constrained by the IPCC AR6 deadline of the end of December 2019. Many modeling groups were in the process of publishing their data to the ESGF, but

since only a few had completed that task, we relied primarily on data sent directly to us. The models included in the figures and analysis were dictated then by those that provided simulation results, which varied by climate variable, to the authors leading this paper. All figures now will include all 17 model simulations for the lig127k experiment published on the CMIP6 ESGF.

2. How was the set of 127k simulations determined? See answer #1 above. The set of simulations that will be included in the revised paper are those that have submitted their lig127k, as well as piControl, simulations to the CMIP6 ESGF. We include a new Table in the Supp Info that details the years analyzed and the DOI references. The CMIP6 database satisfies the publication requirements that all the model simulation data included in the paper is freely and publicly available. As many additional climate variables than those presented in our analyses are available in the CMIP6 database, subsequent, more in-depth topical papers will be possible. In addition, this database includes additional CMIP6 simulations by the modeling groups that could provide interesting past-to-future analyses. With regard to the models in the Scussolini et al. paper, those that are available in the CMIP6 database are now included in our figures and analyses (HadGEM3-GC3.1-LL, IPSL-CR-LR, MPI-ESM1.201p1-LR, NorESM1-f, NESM3).

3. The overlap of models that completed both the lig127 and midHolocene simulations is significant (14 of the 17 models with lig127k simulations also completed midHolocene simulations). We will clearly note where there is not an overlap in the applicable figure legends. Fig. 17 will include all the midHolocene simulations.

4. A related comment concerns comparison to previous last interglacial experiments. We do mention Lunt et al. in the Introduction and will include some additional text where relevant in the revised manuscript. The challenge is that the simulations in the Lunt et al. synthesis are an 'ensemble of opportunity' with different choices for time periods and forcings. PMIP3 and CMIP5 did not include a last interglacial simulation.

[Figure]

The second main comment concerns the robustness of the results and intermodal spread. The standard deviations of ensemble changes in Figure 5 and 9 are intended to demonstrate the variation in the signal between the GCMs. They are the ensemble standard deviation in the temporal mean changes at each location – rather than the ensemble means of the temporal standard deviations. We will add to the multi-model ensemble-average panels stippling where most of the models do not agree on the sign of the change. For Figure 8, we will not only include the correlation coefficients but also a measure of their significance. Intermodel spread is already included in many of the other figures.

The third set of comments concerns the underlying mechanisms and effects of PI biases.

Figure 2 will be deleted in the revised manuscript since with the significant overlap of models with both midHolocene and lig127k simulations, it basically duplicates a similar figure and more extensive discussion in the midHolocene paper. Where relevant, we will refer to the midHolocene paper and include text in the lig127k paper. In place of Figure 2, we will replace the upper panel of Figure 3 with the MAT differences: PI minus observed to more clearly show the PI biases in surface air temperature.

More analyses like Figure 8 would greatly expand the scope of this paper and best left to current (e.g., Kageyama et al., lig127k Arctic sea ice paper) and subsequent (as happened in previous PMIPs) more-detailed, multi-model topical papers and single model papers (e.g. Williams et al., CPD, 2020; O'ishi et al. CPD, 2020). Discussion and references to previously published results will be added to relevant sections.

---

## Author Comment (AC2) · 18 May 2020

Responses to Reviewer #2 cp-2019-174

We would like to thank you for your comments about our manuscript. We believe that the revisions we plan to implement should satisfactorily address your comments.

The first set of similar comments in this review is related to the set of simulations presented in the figures and analysis and included in several queries.

1. Why is HadGEM3 omitted from the sea ice comparison? HadGEM3 was omitted in Figure 8 at the request of the HadGEM3 co-authors. They felt having their results

published online could lead to a paper that they had submitted on the HadGEM3 Arctic sea ice sensitivity not being sent out for review. We will include the HadGEM3 results in the lig127k revised paper.

2. The ACCESS-ESM results in Fig. 16. Indeed, a mistake was found in the dataset provided for the paper. At the time of the submission, as dictated by the IPCC AR6 deadline, many modeling groups were in the process of publishing their data to the ESGF, but since only a few had completed that task, we relied primarily on data sent directly to us. The set of simulations that will be included in the revised paper will be the lig127k and corresponding piControl and midHolocene on the CMIP6 ESGF. A table with the DOIs and analysis years for each model will be included in the Supp Info.

3. The completeness of Table 2. The details will be complete in the revised manuscript. At the time of submission of the lig127k paper, not all groups had provided this information.

The second main comment concerns the representation of model spread and proxy reconstruction uncertainty in Figures 11-13. We agree that these figures do not show the model spread. Indeed, some of the models compare better to the data than others. Nor do these figures show the proxy reconstruction uncertainties, which can be large, as discernable in the Supp Info. The revised paper will address the reviewer's concerns. First, we will include a table in the Supp Info that shows for every core site, the temperature anomaly and its uncertainty from the proxy reconstructions and the temperature anomaly for each model interpolated to the core location. Both annual and seasonal comparisons will be done. Second, a new Figure will be added showing reconstructed temperature anomalies with uncertainty as a function of latitude for each proxy core, and individual model anomalies at each site to show the spread of model estimates. To make readable, separate panels will be drafted regionally and seasonally.

Several more general comments concern the underlying mechanisms. The reviewer raises lots of great questions, in the main and other comments, that would be interest-

ing to address using the lig127k simulations as well as across many of the other PMIP4 and CMIP6 simulations.

- Why might models be getting the incorrect pattern of change in some regions (e.g. is it really all down to missing freshwater or dynamic vegetation)? - What factors might contribute to the relatively large spread in simulated responses? - How robust are the seasonal versus annual reconstructions and which one of these tells us most about the model deficiencies? - Why does the sea-ice loss scale with ECS? - How do models with interactive vegetation or LAI differ from the fixed vegetation models?

This paper is meant to be a more descriptive, similar to the companion CMIP6-PMIP4 papers being published in the PMIP4 Climate of the Past Special Issue. The paper already includes 17 figures. More analyses to answer these questions would greatly expand the scope of this paper and best left to current (e.g., Kageyama et al., lig127k Arctic sea ice paper) and subsequent (as happened in previous PMIPs) more-detailed, multi-model topical papers and single model papers (e.g. Williams et al., CPD, 2020; O'ishi et al. CPD, 2020). Discussion and references to previously published results will be added to relevant sections.

In reply to more specific comments:

Lines 595-604: We agree. The text will be revised.

Lines 550, 552, 555, 562, 585: The text will be revised.

Line 583: The text will be clarified.

Line: 627 and Fig. 8: Now including all lig127k simulations, the correlation between the ECS and simulation of Aug-September lig127k minimum Arctic sea ice extent is -0.6, and is significant at the 95% level. Note that the y-axis in all but the first panel are sea ice area 'anomaly' and are labeled correctly.

Fig. 3, a) We are replacing the upper panel of Figure 3 with the MAT differences: PI minus observed to more clearly show the PI biases in surface air temperature. We will

add the observational uncertainties. b) We prefer to keep the bottom panel as is, i.e. not join the circles. With 17 models now included in this figure, joining the lines would make it less easy to identify individual models (see figure in this reply, note currently for 16 of the models, AWI-ESM-2-1-LR to be added). The models show little spread at low latitudes, a large spread at NH high latitudes.
* * *
[Figure]

[Figure]

**Fig. 1.** Alternate bottom panel for Figure 3

---

## Author Response (AR1)

**Large-scale features of Last Interglacial climate: Results from evaluating the *lig127k* simulations for CMIP6-PMIP4, Otto-Bliesner et al.**

**Overall summary**

The revised paper now includes 17 model simulations for the CMIP6-PMIP4 *lig127k* experiment. Having this more complete set of simulation data allows now for a more comprehensive review of the multi-model mean and range of responses across this large ensemble. The simulations included in this first analysis of the CMIP6-PMIP4 *lig127k* simulations were chosen such that a corresponding CMIP *piControl* simulation had been completed and that the results had been published or will soon be published on the CMIP6 ESGF.

**Reviewer #1**

-> Ensure that the set of climate simulations is complete and that whenever possible the same set is used in all figures and analyses. When this is not the case it should be clearly mentioned. How was the set of 127k simulations determined? Why are for instance some models from the recent publication by Scussolini et al. not included?

The submission of the paper was constrained by the IPCC AR6 deadline of the end of December 2019. Many modeling groups were in the process of publishing their data to the ESGF, but since only a few had completed that task, we relied primarily on data sent directly to us. The models included in the figures and analysis were dictated then by those that provided simulation results, which varied by climate variable, to the authors leading this paper. All figures now will include all 17 model simulations for the lig127k experiment published or soon to be published on the CMIP6 ESGF.

The set of 17 model simulations included in the revised paper are those that have submitted their lig127k, as well as piControl, simulations to the CMIP6 ESGF. We include a new Table in the Supp Info that details the years analyzed and the DOI references. The CMIP6 database satisfies the publication requirements that all the model simulation data included in the paper is freely and publicly available. As many additional climate variables than those presented in our analyses are available in the CMIP6 database, subsequent, more in-depth topical papers will be possible. In addition, this database includes additional CMIP6 simulations by the modeling groups that could provide interesting past-to-future analyses.

With regard to the models in the Scussolini et al. paper, those that are available in the CMIP6 ESGF are now included in our figures and analyses (HadGEM3-GC3.1-LL, IPSL-CM6A-LR, MPI-ESM1-2-LR, NorESM1-F, NESM3).

-> A similar remark for the 6k simulations: ensure that whenever possible the same set is used in all figures and analyses (in figure 17 there are only 5 6k models, is that all of them?) Are the

comparisons of 127k and 6k based only on models that performed both experiments? If an analysis can only be done on a subset of models, make it plausible that the results are not impacted too much by leaving out a substantial number of models.

As noted in response to the previous comment, the IPCC deadline constrained what models and what variables were available to include in the figures. Note that the overlap is significant, with only 3 (ACCESS ESM 1.5, AWI-ESM-2-1-LR, CNRM-CM6-1) of 17 models with lig127k results not having parallel midHolocene results, and 3 (MRI-ESM2, UofT-CCSM-4, BCC-CSM1-2) of 17 midHolocene models not having parallel lig127k results.

Figure 17 now includes the midHolocene simulations with parallel lig127k simulations.

For Figures 15 and 16, we have decided to keep all simulations (14 models overlap) and note in the text/figure legends the non-overlap between the lig127k and midHolocene simulations.

-> Show the robustness of your results and statements by statistical measures. Are the multi-model-mean results significantly different from zero considering the inter-model spread? Are the presented correlations significant?

The standard deviations of ensemble changes in Figure 5 and 9 are intended to demonstrate the variation in the signal between the GCMs. They are the ensemble standard deviation in the temporal mean changes at each location – rather than the ensemble means of the temporal standard deviations. The multi-model ensemble-average panels in these figures now also include stippling where less than 12 of the 17 models agree on the sign of the change, a similar statistic to that shown in Lunt et al. (2013). For Figures 7 and 18, we now include both the correlations and significance values in the text discussing this figure.

-> Indeed PI biases in models can be important, but if these data are presented then a much clearer link to the ensuing 127k and 6k results should be made. Can some of the palaeo results be explained by the PI biases? Are the biases consistent across these different experiments?

Figure 2 has been deleted in the revised manuscript since with the significant overlap of models with both midHolocene and lig127k simulations, it basically duplicates a similar figure and more extensive discussion in the midHolocene paper (Brierley et al., 2020). Where relevant, we refer to the midHolocene paper and include text in the lig127k paper.

The 6 ka results are shown only to contrast the importance of the orbital forcing being smaller than 127 ka. PI biases could be important for explaining the responses of a particular model to the 127 ka and 6 ka orbital forcings. The reasons vary among the models: the representation of clouds, sea ice, ocean circulation, etc, could each be important in some but not all models. This type of detailed attribution is best left to individual model papers or multi-model papers focused on one specific aspect (e.g., Arctic sea ice, Kageyama et al., 2020).

Present more real analyses like in figure 8, not only a description of the results.

The manuscript represents the first results from the CMIP6-PMIP4 lig127k ensemble and therefore we included many comparisons/analyses that subsequent studies should expand on. We acknowledge that the reviewer thought the analysis was too brief, however, the manuscript is an overview for a broad audience and already runs to a number of pages, hence we would not like to add a large amount of additional detail.

We agree with the reviewer that it would be very interesting to understand why large-scale climate features differ among different models. However, this is a very difficult question to answer for the coupled CMIP6 models, and it is beyond the scope of this study to do this thoroughly for all diagnostics.

More analyses like previous Figure 8 (current Figure 7) would greatly expand the scope of this paper and best left to current (e.g., Kageyama et al., lig127k Arctic sea ice paper) and subsequent (as happened in previous PMIPs) more-detailed, multi-model topical papers and single model papers (e.g. Williams et al., CPD, 2020; O'ishi et al. CPD, 2020). Discussion and references to previously published results are added to relevant sections.

-> Hardly any underlying mechanisms are described in the manuscript. Perhaps this is outside of the scope of this manuscript, but it appears to me that many of the mechanisms that can explain the PI, 6k and 127k biases or results have already been discussed elsewhere and references should thus be provided.

The scope of this paper is to be descriptive, providing a multi-model assessment for the IPCC AR6 and overview for future, more mechanism-focused papers. We now include more text and references where relevant and published.

-> Why is no comparison of the 127k results made with previous last interglacial experiments? For instance, Lunt et al. (2012)? Perhaps these experiments did not exactly target 127k, but this could well be of second-order importance for the results of the experiments. Did the model response change from PMIP3 to PMIP4?

We do mention Lunt et al. in the Introduction and now also in Section 3.2. Figure 5 now shows where less than 70% of the 17 models agree on the sign of the change, analogous to Fig. 6 in Lunt et al. The challenge is that the simulations in the Lunt et al. synthesis are an 'ensemble of opportunity' with different choices for time periods and forcings. PMIP3 and CMIP5 did not include a last interglacial simulation.

-> Clarify what this manuscript adds to previous more targeted manuscripts on for instance precipitation (Scussolini et al, 2019) or sea ice (Kageyama et al., 2019).

Section 5.3 and Figure 13 expand on the results of Scussolini et al. (2019) with now the complete set of 17 CMIP6 lig127k simulations. Note Paolo Scussolini is a co-author of this paper and in particular the model-data comparisons for the precipitation responses of the lig127k simulations as compared to the piControl simulations. The more detailed analysis of the

simulation of Arctic sea ice and an evaluation against a new compilation from data can be found in Kageyama et al. (2019). We have added some relevant discussion to the revised manuscript, e.g. sections 3.1, 3.3, 5, and 7.

Reviewer #2

Summary

The authors provide an overview of the lig127k simulations prepared for CMIP6 and PMIP4. They show the main global features of an ensemble of 17 coupled climate models, including the temperature, precipitation and sea-ice response. The article will provide an important reference for the CMIP process and a starting point for further more detailed lig127k analysis. Having said that, the present text could in my opinion better convey what has been learned about the Last Interglacial climate and/or about these climate models. Therefore, I recommend publication once these messages have been brought out, either through more detailed discussion or with further analyses.

Main comments

Several questions feel unanswered at the end of this manuscript, e.g. why might models be getting the incorrect pattern of change in some regions (e.g. is it really all down to missing freshwater or dynamic vegetation)?

What factors might contribute to the relatively large spread in simulated responses?

How robust are the seasonal versus annual reconstructions and which one of these tells us most about the model deficiencies?

Why does the sea-ice loss scale with ECS?

How do models with interactive vegetation or LAI differ from the fixed vegetation models?

Whilst I realise that a full treatment of each of these questions could generate a paper by itself, more discussion would be extremely valuable, especially given that the author list brings together a list of experts for each model and in the palaeoclimate archives.

This paper is meant to be a more descriptive, similar to the companion CMIP6-PMIP4 papers being published in the PMIP4 Climate of the Past Special Issue. The paper already includes 18 figures and 3 tables. More analyses to answer these questions would greatly expand the scope of this paper and best left to current (e.g., Kageyama et al., lig127k Arctic sea ice paper) and subsequent (as happened in previous PMIPs) more-detailed, multi-model topical papers and single model papers (e.g. Williams et al., CPD, 2020; O'ishi et al. CPD, 2020). Discussion and references to previously published results is added to relevant sections.

All good questions that would be excellent for future, more focused paper to address, but beyond the scope of this paper. This paper provides a multi-model assessment for the IPCC AR6 and overview for to-be-written, more mechanism-focused papers. This can best be done in single model papers with the experts of a specific component model. More difficult to do in more focused, multi-model papers. Correlations are possible but mechanisms not so easily assessed in the complex multi-component CMIP6 models.

Other comments

If I understand correctly, you are showing that models with higher ECS also show stronger Arctic sea-ice loss in LIG simulations. This despite lower GHG levels in the lig127k simulations. I think this is interesting, but requires more analysis.

Yes, very interesting. This figure is now included in Section 7 with an expanded discussion. Others (e.g. Schmidt et al, 2014) have proposed that simulated and proxy MH ice extent anomalies might be able to be used to estimate the likely loss in future projections. Yoshimori and Suzuki (2019) documented local Arctic feedbacks as important common contributors in the CMIP5 MH and RCP4.5 simulations of 10 climate models. Kageyama et al. (2020) illustrate an almost linear relationship between the simulations of Arctic summer sea ice in the lig127k and 1pctCo2 simulations by the CMIP6 models. Most recently, Guarino et al. (2020) suggest that the high ECS and summer ice-free Arctic in the HadGEM3.GC3.1-LL is reasonable in light of its improved simulation of LIG Arctic summer temperatures as compared to HadCM3.

It feels incomplete to omit HadGEM3 from the sea-ice ECS comparison, especially as this model has the largest polar warming in the ensemble. Please can you include this in the analysis?

We agree. HadGEM3 is now included in the ECS-summer sea ice panel of Figure 18. It was omitted in the submission by request of the HadGEM3 lig127k authors due to a conflict with another paper in review at Nature Climate Change. That paper has now been published (Guarino et al., 2020).

Lines 595-604: I don't really agree with the main point here. None of the dynamic vegetation models in PMIP2 or CMIP5 showed an adequate precipitation response in North Africa, so why should this differ for the last interglacial?

We agree. This sentence has been deleted. In addition, we added two sentences at the end of Section 3.4 that the 4 models that include interactive vegetation fall within the spread of the models with prescribed vegetation for the three metrics and 7 monsoon regions.

Table 2: The details of the models included needs to be completed before publication. What is the HadGEM3 ECS and how were vegetation and aerosols treated in FGOALS, GISS-E2-1-G and NESM3 and CNRM?

Table is now complete.

Fig. 11-13: I think some assessment of the model uncertainty and paleo reconstruction uncertainty is required. It's not clear from these figures whether the multi-model mean is good but the individual models are biased etc.

We agree that these figures do not show the model spread. Indeed, some of the models compare better to the data than others. Nor do these figures show the proxy reconstruction uncertainties, which can be large. Indeed, a challenge is to show both in a figure. Section 4 has been revised to address the data reconstructions and their uncertainties. The revised paper now includes new SI tables. Tables in the SI detail for every core site, the temperature anomaly and its uncertainty from the proxy reconstructions, and the temperature anomaly for each model interpolated to the core location. For a Figure, we settled on the new Figure 12 showing reconstructed temperature anomalies with uncertainty as a function of latitude for the proxy sites, and individual model anomalies at each site to show the spread of model estimates. To make readable, separate panels show the most interesting regional and seasonal comparisons. The tables in the SI provide the details for each model. Figure 12 complements Figures 10 and 12, which show the MMM versus the proxy reconstructions.

Technical corrections

Line 550: "though with significant differences among the models". Line 552: "but with substantial differences among the ensemble" Line 555: "though again with a large spread across the model ensemble" Line 562: "The spread across the multi-model ensemble is particularly large for the North African monsoon" Line 585: "However, the model spread is large" Please quantify these.

These sentences have been rewritten to clarify their meanings and provide quantified measures.

Line 583: "The most consistent picture from the temperature proxies representing annual conditions is warmer LIG temperatures over Greenland and Antarctica" What do you mean here? Consistent between model simulations and reconstructions, or consistent within the reconstructions?

The text has been clarified.

Line 627: "There appears to be a clear relationship between the ECS of each model and its simulation of August-September lig127k minimum Arctic sea ice extent"

I'm not sure this is accurate. The comparison in the first panel of figure 8 (which needs to be properly labelled) is moderate at best, but perhaps I have misunderstood the plots as the labels are inconsistent with the caption. The r2 would be useful here.

This Figure have been moved to Section 7 and the discussion expanded. The correlation is -0.61 and is significant at the 0.05 level (ncl function: rtest).

Figures

Fig. 3: a) I can see why you have offset the models and observations in the latitudinal direction, but this could lead to confusion. Would this plot not work better as anomalies? Mostly what we see here is that the models capture the latitudinal temperature gradient. Additionally what is the uncertainty on the observations?

Fig. 3 is now Fig. 2 in the revised manuscript. In this figure, as well as subsequent showing individual models, we use different symbols, colors, line types to better distinguish among the 17 models.

Thank you for the suggestion. Fig. 2a now shows the anomalies, PI-obs. The +/- 1 standard deviation for the observations has been added to the figure.

b) Please can you join the circles in the lower plot with lines, so that we can see the integrated latitudinal change in each model separately. For example, this might show that a model is the warmest at high-latitudes but is in the middle of the ensemble elsewhere?

Thank you for the suggestion. Fig. 2b now shows the zonal averages for each model's lig127k minus piControl surface air temperature change.

Fig. 8: the y-axes are incorrectly labelled in all but the first panel. There are also grey lines between some of the panels.

This figure (now Fig. 7) has been revised.

Fig. 16: ACCESS-ESM appears to show close to no change in these figure panels - can you double check this?

The results sent by the ACCESS-ESM group last December had an error. Both were PI. This has now been corrected. This shows the importance of using the CMIP6 database. The DOIs and years analyzed for each of the model simulations are included in a new table, Table S2 in the Supplementary Information.

[revised manuscript text omitted]

**Commented [BO1]:** This table has been revised to:
1) Added models with lig127k available in ESGF. Adopted official CMIP6 model names. DOIs for data access given in Table S2. LOVECLIM model no longer included, with concurrence of model authors, as data will not be published in the CMIP6 ESGF.
2) Completed details and references for models.
3) ECS for all models now consistently calculated (see footnote)

| | | | | |
|---|---|---|---|---|
| MPI-ESM1-2-LR | AWI
MPI-Met | Giorgetta et al., 2013 | 3.0°C | Scussolini et al., 2019

Interactive vegetation
Prescribed aerosols |
| NESM3 | NUIST | Cao et al. (2018) | 4.7°C | Interactive vegetation
Prescribed aerosols |
| NorESM1-F | Norwegian Climate
Centre, NCC | Guo et al., GMD, 2019 | 2.3°C | Prescribed vegetation and aerosols as PI |
| NorESM2-LM | Norwegian Climate
Centre, NCC | Seland et al., 2020 | 2.5°C | Prescribed vegetation and aerosols as PI |

1780

[1]ECS uses the Gregory method from a 150-year run of an instantaneously quadrupled $CO_2$ simulation (Meehl et al., 2020, Wyser et al., 2020)

1785

**Table 3. Metrics for surface air temperature change (°C ) for CMIP6-PMIP4 *lig127k* simulations**

**Commented [BO2]:**
This new table quantifies surface temperature changes.

| Climate Model | Global | Global Land | Global Ocean | NH | NH Land | NH Ocean | SH | SH Land | SH Ocean | NH Meridional Gradient[1] | SH Meridional Gradient[2] |
|---|---|---|---|---|---|---|---|---|---|---|---|
| ACCESS-ESM1-5 | 0.33 | 0.42 | 0.29 | 0.43 | 0.34 | 0.48 | 0.23 | 0.58 | -0.05 | 1.61 | 1.89 |
| AWI-ESM-1-1-LR | -0.25 | -0.47 | -0.16 | -0.55 | -0.81 | -0.37 | 0.04 | 0.25 | -0.08 | 0.38 | 0.86 |
| AWI-ESM-2-1-LR | -0.20 | -0.34 | -0.14 | -0.39 | -0.59 | -0.25 | -0.01 | 0.20 | -0.14 | 0.8 | 0.78 |
| CESM2 | -0.11 | -0.16 | -0.09 | -0.22 | -0.31 | -0.16 | 0.00 | 0.18 | -0.08 | 1.02 | 0.47 |
| CNRM-CM6-1 | 0.4 | 0.39 | 0.4 | 0.33 | 0.15 | 0.46 | 0.46 | 0.89 | 0.26 | 1.21 | 0.55 |
| EC-Earth3-LR | 0.45 | 0.71 | 0.34 | 0.99 | 0.92 | 1.03 | -0.07 | 0.32 | -0.17 | 3.94 | 0 |
| FGOALS-f3-L | -0.48 | -0.57 | -0.44 | -0.60 | -0.77 | -0.48 | -0.37 | -0.16 | -0.35 | 0.3 | -0.28 |
| FGOALS-g3 | 0.38 | 0.6 | 0.29 | 0.38 | 0.51 | 0.29 | 0.48 | 0.89 | 0.24 | 2.42 | 1.14 |
| GISS-E2-1-G | -0.12 | -0.1 | -0.13 | -0.07 | -0.17 | 0.00 | -0.18 | 0.06 | -0.20 | 1.59 | -0.11 |
| HadGEM3-GC31-LL | 0.56 | 0.71 | 0.49 | 0.89 | 0.76 | 0.97 | 0.22 | 0.62 | 0.08 | 3.08 | 0.37 |
| INM-CM4-8 | -0.2 | -0.3 | -0.15 | -0.30 | -0.54 | -0.14 | -0.09 | 0.20 | -0.12 | 0.45 | -0.23 |
| IPSL-CM6A-LR | -0.29 | -0.3 | -0.29 | -0.29 | -0.43 | -0.19 | -0.30 | -0.03 | -0.31 | 0.89 | -0.02 |
| MIROC-ES2L | -0.4 | -0.55 | -0.33 | -0.52 | -0.73 | -0.38 | -0.26 | -0.12 | -0.29 | 0.92 | 0.55 |
| MPI-ESM1-2-LR | -0.12 | -0.24 | -0.07 | -0.33 | -0.54 | -0.19 | 0.10 | 0.42 | -0.05 | 0.95 | 0.83 |
| NESM3 | 0.07 | -0.02 | 0.11 | -0.25 | -0.43 | -0.12 | 0.39 | 0.86 | 0.22 | 0.83 | 0.57 |
| NorESM1-F | -0.24 | -0.35 | -0.2 | -0.33 | -0.55 | -0.18 | -0.15 | 0.08 | -0.21 | 0.59 | 0.24 |
| NorESM2-LM | -0.11 | -0.04 | -0.14 | -0.13 | -0.13 | -0.12 | -0.09 | 0.16 | -0.16 | 0.69 | 0.39 |
| | | | | | | | | | | | |
| **Mean** | **-0.02** | **-0.04** | **-0.01** | **-0.06** | **-0.20** | **0.04** | **0.02** | **0.32** | **-0.08** | **1.27** | **0.47** |
| **Std dev** | **0.32** | **0.44** | **0.28** | **0.48** | **0.54** | **0.45** | **0.26** | **0.34** | **0.19** | **1.00** | **0.55** |
| **Max** | **0.56** | **0.71** | **0.49** | **0.99** | **0.92** | **1.03** | **0.48** | **0.89** | **0.26** | **3.94** | **1.89** |
| **Min** | **-0.48** | **-0.57** | **-0.44** | **-0.60** | **-0.81** | **-0.48** | **-0.37** | **-0.16** | **-0.35** | **0.30** | **-0.28** |

1790

[1] 60-90N minus 0-30N
[2] 60-90S minus 0-30S

| Page 7: [1] Deleted | Bette Otto-Bliesner | 8/22/20 1:24:00 PM |
| Page 7: [2] Deleted | Bette Otto-Bliesner | 7/12/20 2:54:00 PM |
| Page 7: [3] Deleted | Bette Otto-Bliesner | 7/16/20 12:59:00 PM |
| Page 7: [4] Deleted | Bette Otto-Bliesner | 7/16/20 4:47:00 PM |
| Page 7: [5] Deleted | Bette Otto-Bliesner | 8/22/20 3:32:00 PM |

---

## Referee Report (RR1)

The revised manuscript is much improved compared to the previous version and is in my view nearly ready for publication.
I have two minor comments that I feel should be addressed, and a number of technical points.

Minor comments.

In the abstract the authors mention that this work enhances our confidence in future projections of the stability of the Greenland Ice Sheet. I don't see any model or proxy evidence presented in the manuscript to substantiate such a claim.

In the introduction it is said that previous LIG intercomparisons pointed to cryosphere or ocean circulation feedbacks to explain intermodel differences, but that those simulations also differed in the experimental design, making it difficult to draw conclusions. This is indeed true, but now that we have a large set of simulations that do include the same experimental design, what can the authors at to this discussion? Were the previously published inter-model differences the result of flaws in the experimental design, or are feedbacks responsible? The authors touch upon this topic at various points in the manuscript, but the link to previous literature can be made more explicit.

Technical comments:

Line 294: do we know that changes in the size of the Antarctic ice sheet during the Last Interglacial impacted the meridional temperature gradient?

Line 131: Arctic sea ice?

Lines 206-215: It would be good to make it more explicit that you are not talking about the lig127k simulations here, but their corresponding PI control simulations. The way it is worded now might be confusing to the reader.

Line 272: greenhous-gas concentrations

Line 306: (2020)

Lines 1192: typo subscript instead of superscript.

Line 550: MMM

Line 533: correct sentence

Line 534: changes that are

---

## Author Response (AR2)

Responses (blue type) to Reviewer comments

Reviewer #1

Line 82: "These equilibrium simulations are designed to examine the impact of changes in the Earth's orbit on the latitudinal and seasonal distribution of incoming solar radiation (insolation) at times when ..."
I think what you mean is:
"These equilibrium simulations are designed to examine the impact of changes in the Earth's orbit and hence the latitudinal and seasonal distribution of incoming solar radiation (insolation) on Earth's climate at times when ..."
Yes, much better phrasing. Rewritten as suggested

Line 159: delete the underscore after the
Corrected

Line 292: replace "for" with "from"?
Replaced

Line 306: Missing ")" after "2020"?
Corrected

Line 685: Floating sentence?
Corrected

Figure 7: The legend is incorrect (top and bottom are mixed up).
Figure has been redrafted so that panels match the legend

Guarino et al (2020) is missing from the reference list.
Added

Table 2: can you clarify the difference between "Prescribed vegetation and aerosols" and "Prescribed vegetation and aerosols as PI"?
Except as noted, the models adopted their PI vegetation and aerosols. A footnote has been added to the table.

Figure 12: This is very detailed and I think useful. Can you ensure the legend is less blurred?
Legend redone to be sharper

Figure 13 a is also quite blurred.
Figure has been improved

Figure 16: dashed zero line is missing.
Dashed zero line added

Can you change km2 so that the 2 is superscript please, and similar for 14C.
This occurred during the conversion from Word document to a pdf.

Reviewer #2

Minor comments.
In the abstract the authors mention that this work enhances our confidence in future projections of the stability of the Greenland Ice Sheet. I don't see any model or proxy evidence presented in the

manuscript to substantiate such a claim.

Deleted this claim.

In the introduction it is said that previous LIG intercomparisons pointed to cryosphere or ocean circulation feedbacks to explain intermodel differences, but that those simulations also differed in the experimental design, making it difficult to draw conclusions. This is indeed true, but now that we have a large set of simulations that do include the same experimental design, what can the authors at to this discussion? Were the previously published inter-model differences the result of flaws in the experimental design, or are feedbacks responsible? The authors touch upon this topic at various points in the manuscript, but the link to previous literature can be made more explicit.

The reviewer is correct. We have added a sentence to the Discussion (lines 630-634) acknowledging that even with a common experimental protocol, there is a large spread across the models in simulating Arctic sea ice. We reference Kageyama et al., CP, 2020, "A multi-model CMIP6 study of Arctic sea ice at 127ka: Sea ice data compilation and model differences", which includes an analysis of differences in the atmospheric energy budgets, ocean circulations, and atmospheric circulations among the *lig127k* simulations. They conclude that it is not possible to generalize the reasons for this spread across the model ensemble.

Technical comments:
Line 294: do we know that changes in the size of the Antarctic ice sheet during the Last Interglacial impacted the meridional temperature gradient?

A sentence and citations added on how the size of the Antarctic ice sheet may have impacted the SH latitudinal gradient.

Line 131: Arctic sea ice?

Yes, corrected.

Lines 206-215: It would be good to make it more explicit that you are not talking about the lig127k simulations here, but their corresponding PI control simulations. The way it is worded now might be confusing to the reader.

Yes, corrected.

Line 272: greenhous-gas concentrations

Corrected

Line 306: (2020)

Corrected

Lines 1192: typo subscript instead of superscript.

This occurred during the conversion from Word document to a pdf.

Line 550: MMM

Corrected

Line 533: correct sentence

Corrected

Line 534: changes that are

We have rewritten this statement to better describes the results shown in Fig. 12d and Table S5.

[revised manuscript text omitted]